# Prolonged β-adrenergic stimulation disperses ryanodine receptor clusters in cardiomyocytes and has implications for heart failure

Xin Shen[1,2]*, Jonas van den Brink[3], Anna Bergan-Dahl[1,2], Terje R Kolstad[1,2], Einar S Norden[1,2], Yufeng Hou[1,2], Martin Laasmaa[1,2], Yuriana Aguilar-Sanchez[4,5], Ann P Quick[4,5], Emil KS Espe[1,2], Ivar Sjaastad[1,2], Xander HT Wehrens[4,5], Andrew G Edwards[3,6], Christian Soeller[7], William E Louch[1,2]*

[1]Institute for Experimental Medical Research, Oslo University Hospital and University of Oslo, Oslo, Norway; [2]K.G. Jebsen Centre for Cardiac Research, University of Oslo, Oslo, Norway; [3]Simula Research Laboratory, Lysaker, Norway; [4]Section of Cardiology, Departments of Medicine and Pediatrics, Baylor College of Medicine, Houston, United States; [5]Department of Molecular Physiology & Biophysics, Cardiovascular Research Institute, Baylor College of Medicine, Houston, United States; [6]Department of Pharmacology, UC Davis, Davis, United States; [7]Department of Physiology, University of Bern, Bern, Switzerland

*For correspondence:
xin.shen@medisin.uio.no (XS);
w.e.louch@medisin.uio.no (WEL)

Competing interest: The authors declare that no competing interests exist.

**Abstract** Ryanodine receptors (RyRs) exhibit dynamic arrangements in cardiomyocytes, and we previously showed that 'dispersion' of RyR clusters disrupts $Ca^{2+}$ homeostasis during heart failure (HF) (Kolstad et al., eLife, 2018). Here, we investigated whether prolonged β-adrenergic stimulation, a hallmark of HF, promotes RyR cluster dispersion and examined the underlying mechanisms. We observed that treatment of healthy rat cardiomyocytes with isoproterenol for 1 hr triggered progressive fragmentation of RyR clusters. Pharmacological inhibition of $Ca^{2+}$/calmodulin-dependent protein kinase II (CaMKII) reversed these effects, while cluster dispersion was reproduced by specific activation of CaMKII, and in mice with constitutively active Ser2814-RyR. A similar role of protein kinase A (PKA) in promoting RyR cluster fragmentation was established by employing PKA activation or inhibition. Progressive cluster dispersion was linked to declining $Ca^{2+}$ spark fidelity and magnitude, and slowed release kinetics from $Ca^{2+}$ propagation between more numerous RyR clusters. In healthy cells, this served to dampen the stimulatory actions of β-adrenergic stimulation over the longer term and protect against pro-arrhythmic $Ca^{2+}$ waves. However, during HF, RyR dispersion was linked to impaired $Ca^{2+}$ release. Thus, RyR localization and function are intimately linked via channel phosphorylation by both CaMKII and PKA, which, while finely tuned in healthy cardiomyocytes, underlies impaired cardiac function during pathology.

## Editor's evaluation

This article is of fundamental interest to our understanding of heart failure and the cardiac contraction process. It applies super-resolution imaging and functional calcium imaging to healthy and failing cardiac cells and combines these with quantitative modeling of the resulting data. The outcome bearing on receptor distribution is a good example of exploiting quantitative super-resolved data in combination with other techniques to gain real insight into a biological problem.

## Introduction

In cardiomyocytes, the processes underlying initiation of contraction are well described, at least at the macroscale. Sarcolemmal depolarization triggers an influx of $Ca^{2+}$ through voltage-gated L-type $Ca^{2+}$ channels (LTCCs) within t-tubules, which in turn elicits a much larger release of $Ca^{2+}$ into the cytosol via ryanodine receptors (RyRs) in the sarcoplasmic reticulum (SR). This process of $Ca^{2+}$-induced $Ca^{2+}$ release (CICR) depends on the precise organization of LTCCs and RyRs within dyadic junctions of the two membranes, with LTCCs located in nanoscale apposition to RyRs across the dyadic cleft (*Bers, 2001*). Intrinsically, RyRs are organized into discrete clusters, and these arrangements are thought to be critical in defining channel function. Indeed, RyR clusters located within close proximity are proposed to cooperatively operate as $Ca^{2+}$ release units (CRUs) to generate $Ca^{2+}$ sparks (*Baddeley et al., 2009*; *Sobie et al., 2006*), the elementary units of SR $Ca^{2+}$ release (*Cheng et al., 1993*). Recent advances in high-resolution imaging techniques such as 3D dSTORM (*Shen et al., 2019*), DNA-PAINT (*Jayasinghe et al., 2018*), and electron tomography *Asghari et al., 2014* have created an opportunity to understand these dyadic arrangements in unprecedented detail.

Contractile dysfunction and arrhythmogenesis are hallmarks of heart failure (HF), and considerable data have linked these phenomena to t-tubule disruption in this condition (*Fowler et al., 2020*; *Louch et al., 2006*; *Orchard et al., 2013*). Recently, we reported that pathological changes occur also on the other side of the dyad, as we observed 'dispersion' of RyR clusters in failing cardiomyocytes. This RyR rearrangement specifically included fragmentation of RyR groupings into more numerous, smaller clusters, without any change in overall channel number (*Kolstad et al., 2018*). RyR dispersion was critically linked to low-fidelity spark generation. Furthermore, when $Ca^{2+}$ release was successfully triggered by a CRU, propagation of $Ca^{2+}$ between multiple clusters generated sparks with slow kinetics, and the overall $Ca^{2+}$ transient was desynchronized (*Kolstad et al., 2018*). Similar fragmentation of CRUs has been reported in a sheep model of persistent atrial fibrillation (*Macquaide et al., 2015*) and in monocrotaline-induced right ventricular failure in rats (*Sheard et al., 2019*). Thus, accumulating evidence indicates that RyR mislocalization, and ensuing dysfunction, is a key contributor to pathophysiology.

What drives changes in RyR organization in diseased cardiomyocytes? Previous work has indicated that nanoscale RyR positioning is fine-tuned by phosphorylation of the channels, at least acutely (*Asghari et al., 2014*; *Asghari et al., 2020*), and that both $Ca^{2+}$/calmodulin-dependent protein kinase II (CaMKII)- and protein kinase A (PKA)-dependent phosphorylation of RyR disrupt its function during HF (*Ling et al., 2009*; *Marx et al., 2000*; *Zhang et al., 2003*). We presently hypothesized that these two phenomena are intimately linked. We demonstrate that prolonged β-adrenergic stimulation, as is well known to occur in HF, promotes RyR dispersion via both CaMKII- and PKA-dependent phosphorylation of the channel. This RyR reorganization and sensitization are associated with a time-dependent increase in $Ca^{2+}$ leak, slowing of $Ca^{2+}$ spark kinetics, and reduced $Ca^{2+}$ transient magnitude. In healthy cells, these actions appear to be aimed at gradually countering the stimulatory effects of increased RyR phosphorylation and $Ca^{2+}$ sensitivity. Indeed, we observed that sufficiently dispersed RyRs inhibit the development of $Ca^{2+}$ waves that underlie arrhythmia. Conversely, blocking hyperphosphorylation of RyRs reverses channel dispersion in HF cells and associated impairment of $Ca^{2+}$ release, providing new insight into the protective mechanisms of β-blockade in this disease.

## Results

### Prolonged β-adrenergic receptor (β-AR) activation causes RyR cluster dispersion

Using 3D dSTORM imaging, we investigated the effects of long-term phosphorylation on RyR organization in cardiomyocytes. We specifically hypothesized that phosphorylation would fragment or 'disperse' RyR clusters in a time-dependent manner. Numerically, such breaking apart of RyR groupings is expected to be reflected by a reduction in the average number of RyRs contained in each cluster, as illustrated in (*Figure 1—figure supplement 1*). This change would be accompanied by a concomitant increase in the total number of clusters as there are now more numerous, smaller clusters in a given 3D space. If marked RyR cluster dispersion occurs, the total number of RyRs contained in a CRU may also be reduced as clusters are no longer located in close enough proximity to be grouped in functional release units (edge-to-edge distances ≤ 100 nm; *Figure 1—figure supplement*

*1*). Notably, if RyR dispersion occurs without the addition or loss of channels, then the total number of RyRs detected by dSTORM imaging should remain unchanged.

We compared RyR arrangements in normal rat ventricular cardiomyocytes to those treated with isoproterenol (100 nM) for varying lengths of time (10, 30, or 60 min). For the control group, cells were kept for 60 min following isolation prior to fixation for imaging. 3D dSTORM reconstructions revealed progressive RyR cluster dispersion as the duration of isoproterenol application increased (*Figure 1a*). Indeed, following a brief, 10 min bout of isoproterenol exposure, we detected a slight, but non-significant, change in measured cluster parameters. However, after 30 min of isoproterenol treatment, RyR cluster size significantly decreased (*Figure 1b*), while cluster density increased (*Figure 1d*). By 1 hr of β-AR activation, continued RyR rearrangement was sufficient to reduce CRU size as fewer RyRs remained within clusters located ≤ 100 nm from their neighbors (*Figure 1c*, see enlargements in *Figure 1a* for representative images of CRUs). Since overall RyR density was unchanged by isoproterenol treatment (*Figure 1e*), the detected differences in RyR organization did not result from channel degradation, but rather indicate progressive dispersion of RyR clusters into more numerous, smaller groupings.

We next performed 3D correlative imaging of RyRs and the t-tubular network to distinguish between dyadic and non-dyadic ('orphaned') RyRs (*Figure 1f*). Based on these analyses, the majority of RyR clusters were classified as dyadic, and this proportion was observed to be similar under control and isoproterenol treatment conditions (83 ± 1.6% and 84 ± 1.0%, respectively; *Figure 1g*). Importantly, RyR cluster dispersion was found to be restricted to RyRs within dyads (control: 9.0 ± 0.5 vs. 60 min isoproterenol: 6.7 ± 0.4 RyRs/cluster, *Figure 1h*). It should be noted that the above analyses were performed on RyR clusters within the cell interior. Interestingly, the organization of surface RyR clusters associated with the cardiomyocyte sarcolemma was largely unaffected by β-AR activation save for a slight increase (p=0.092) in cluster size following 10 min isoproterenol treatment (*Figure 1—figure supplement 2*), suggesting that RyR dispersion is a phenomenon restricted to internal, dyadic RyRs. We thus focus only on interior RyR clusters for the remainder of the study.

## CaMKII activation mediates isoproterenol-induced RyR cluster dispersion

In cardiomyocytes, it is well established that a substantial portion of β-AR signaling is mediated via CaMKII (*Grimm and Brown, 2010*) and includes CaMKII-mediated phosphorylation of RyRs (*Ferrero et al., 2007*; *Grimm et al., 2015*). Indeed, using flow cytometry analysis, we observed increased RyR phosphorylation at the CaMKII site ser-2814 that was particularly prominent during early stages of isoproterenol treatment (*Figure 2—figure supplement 1a–c*). We therefore hypothesized that isoproterenol-induced cluster dispersion is, at least in part, driven by the actions of CaMKII. To this end, we first tested the effects of the CaMKII competitive inhibitor autocamtide-2-related inhibitory peptide (AIP, 2 μM) following an initial hour-long isoproterenol incubation (i.e., 1 hr isoproterenol followed by 1 hr isoproterenol + AIP). We found that the addition of AIP markedly reversed cluster dispersion (*Figure 2a*). This was reflected by a significant increase in both RyR cluster size (*Figure 2d*) and CRU size (*Figure 2e*), as well as a significant reduction in RyR cluster density (*Figure 2f*). In control experiments, the longer 2 hr treatment period with isoproterenol alone did not alter RyR configuration beyond values observed at the 1 hr time point (*Figure 2—figure supplement 2a and b*). Direct activation of CaMKII through the Epac2 pathway using 8-CPT-cAMP (10 μM) (*Pereira et al., 2013*) also consistently reproduced the RyR cluster dispersion observed during isoproterenol stimulation (*Figure 2b*, quantification in *Figure 2d–f*). RyR dispersion was again reversed by the addition of AIP post-treatment in these cells. To further corroborate our results, we indirectly stimulated CaMKII activity by applying a low concentration of caffeine (0.5 mM) to cardiomyocytes to increase spontaneous $Ca^{2+}$ release from the SR (*Øyehaug et al., 2013*). We found that caffeine application gave rise to significant reductions in both RyR cluster and CRU size, as well as a significant increase in cluster density (*Figure 2c*, quantification in *Figure 2d–f*). Again, these changes were reversed when AIP was administered.

While these findings suggest that isoproterenol-induced cluster dispersion is dependent on CaMKII, it remained unclear whether these effects can be directly attributed to its phosphorylation of RyRs. To address this issue, we examined RyR cluster characteristics in cardiomyocytes isolated from transgenic mice in which the CaMKII phosphorylation site on RyR is either constitutively active

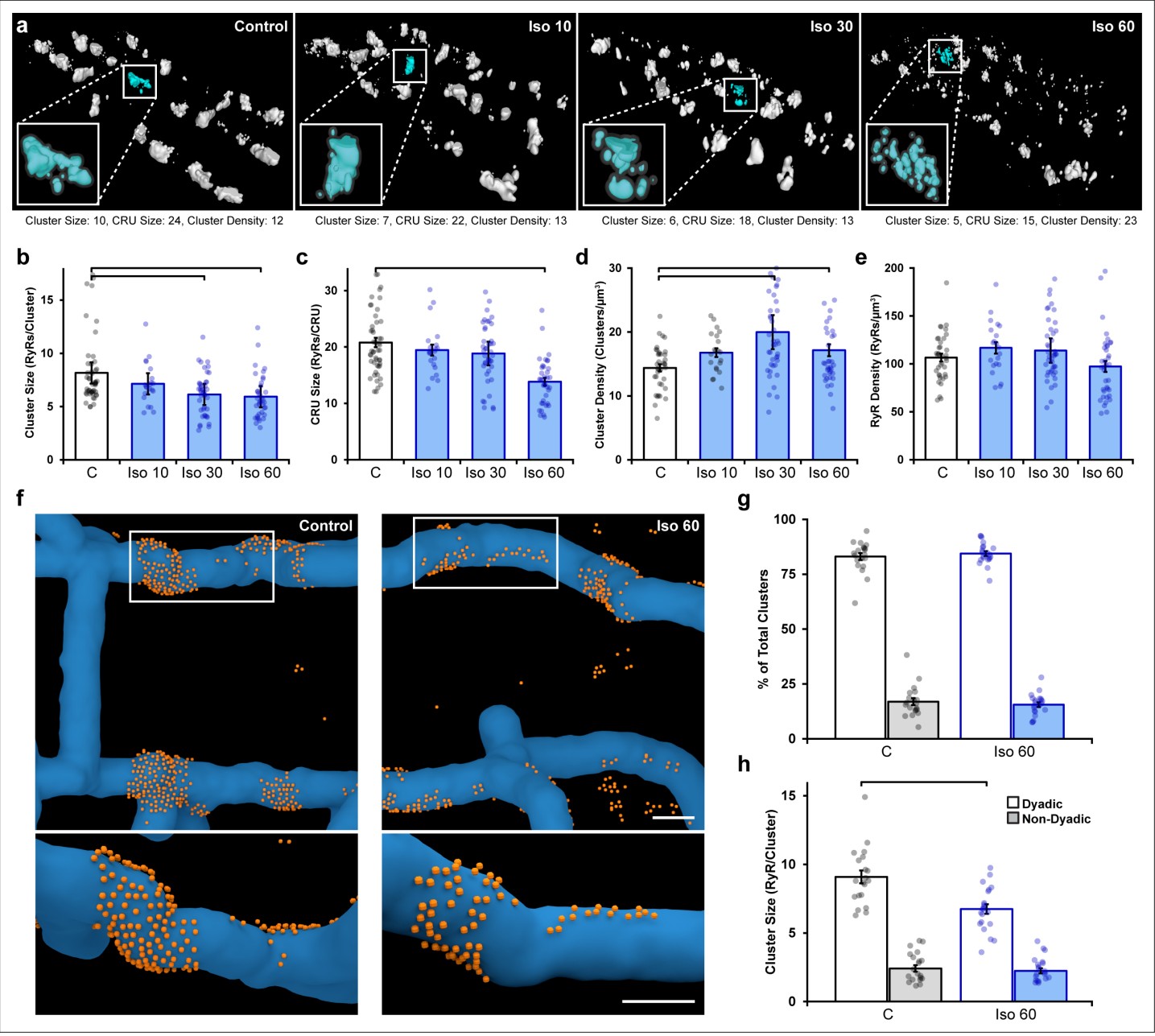

**Figure 1.** Prolonged β-adrenergic receptor (β-AR) activation disperses ryanodine receptor (RyR) clusters. (**a**) Representative reconstructions of interior RyR clusters based on 3D dSTORM in isolated rat cardiomyocytes (clusters were cropped from a region measuring 5 x 8 x 0.6 μm). Control conditions are compared with isoproterenol treatment (100 nM) of varying duration (10, 30, and 60 min). The insets depict single Ca²⁺ release units (CRUs), which encompass RyR clusters with edge-to-edge distances ≤ 100 nm. (**b–d**) Quantification of the dSTORM data revealed progressive dispersion of RyR clusters during isoproterenol treatment, as indicated by measurements of RyR cluster size, CRU size, and cluster density. (**e**) RyR density was unchanged (control: $n_{cells}$ = 50, $n_{hearts}$ = 6; Iso 10: $n_{cells}$ = 21, $n_{hearts}$ = 3; Iso 30: $n_{cells}$ = 43, $n_{hearts}$ = 4; Iso 60: $n_{cells}$ = 37, $n_{hearts}$ = 5). (**f**) Correlative imaging of the t-tubular network (confocal microscopy) and RyRs (dSTORM) was employed to create 3D reconstructions of dyadic and non-dyadic CRUs (t-tubules: blue; RyRs: orange). Scale bars: 250 nm. (**g**) Similar proportions of clusters were characterized as dyadic under control conditions and following 60 min isoproterenol. (**h**) Cluster size measurements indicated that dispersion during isoproterenol occurred exclusively within dyads (control: $n_{cells}$ = 20, $n_{hearts}$ = 3; Iso 60: $n_{cells}$ = 21, $n_{hearts}$ = 3). The bar charts present mean measurements ± SEM, with superimposed data points representing averaged values from each cardiomyocyte. Statistical significance (p<0.05) between groups is indicated by a comparison bar.

The online version of this article includes the following figure supplement(s) for figure 1:

**Figure supplement 1.** Schematic illustrating hallmarks of ryanodine receptor (RyR) cluster dispersion.

*Figure 1 continued on next page*

**Figure supplement 2.** β-Adrenergic receptor (β-AR) stimulation does not significantly influence ryanodine receptor (RyR) organization on the cell surface.

**Figure supplement 3.** Method for correlative imaging of ryanodine receptors (RyRs) and t-tubules for dyad reconstruction.

(S2814D) or genetically ablated (S2814A). We found that the RyR arrangement in the phosphomimetic S2814D mutant cells *at baseline* resembled the fragmented RyR organization observed in wildtype (WT) animals after isoproterenol treatment (*Figure 2g and h*). Indeed, cluster size (*Figure 2j*), CRU size (*Figure 2k*), and cluster density (*Figure 2l*) were all similar in WT + Iso and S2814D cardiomyocytes, and significantly different from untreated WT cells. Furthermore, application of isoproterenol to S2814D cardiomyocytes did not lead to significant additional RyR dispersion, and only tendencies toward smaller cluster and CRU sizes (*Figure 2h and j–l*). In contrast, cardiomyocytes from non-phosphorylatable S2814A mutant mice exhibited similar RyR organization as in WT mice at baseline (*Figure 2i*). Administering isoproterenol to this group promoted a significant reduction in cluster size (*Figure 2j*) and increased cluster density (*Figure 2l*). These findings mirror observations in rat cardiomyocytes, where CaMKII inhibition did not fully reverse RyR dispersion in several experiments (*Figure 2d–f*). Taken together, these results support that CaMKII-dependent RyR phosphorylation during prolonged β-AR stimulation is a key driver for cluster dispersion, but that there are likely additional contributing mechanisms.

## PKA-dependent phosphorylation also contributes to RyR cluster dispersion during β-AR activation

Following β-AR stimulation, increased cyclic AMP levels lead to PKA-dependent phosphorylation of the RyR at ser-2808 (*Marx et al., 2000*; *Figure 2—figure supplement 1d*). Based on the above results, we hypothesized that in parallel to CaMKII-mediated RyR phosphorylation, activation of PKA following isoproterenol application also contributes to RyR cluster dispersion. Indeed, β-AR-induced cluster dispersion was reversed by the addition of the PKA inhibitor H89 (*Figure 3a*), as reflected by significantly increased cluster size (*Figure 3c*) and CRU size (*Figure 3d*), and reduced cluster density (*Figure 3e*). As in experiments investigating the effects of CaMKII inhibition (*Figure 2*), it should be noted that PKA inhibition did not fully restore the RyR configuration at the CRU level. Indeed, simultaneous treatment of cardiomyocytes with AIP and H89 resulted in further reversal of the effects of isoproterenol on CRU size, beyond those observed with either agent alone (*Figure 3—figure supplement 1a–c*). These observations are consistent with summative roles of PKA and CaMKII activation in promoting RyR dispersion.

To further validate an involvement of PKA, we quantified RyR arrangement following application of the selective PKA activator 6MB-cAMP (6MB, 100 µM) (*Christensen et al., 2003*; *Szabo-Fresnais et al., 2010*). As illustrated in *Figure 3b*, 6MB-cAMP treatment prompted RyR dispersion comparable to that elicited by isoproterenol. These changes were again reversible by co-incubating 6MB with H89 following an initial treatment period with 6MB alone (*Figure 3c–e*). Importantly, 6MB induced RyR dispersion in S2814A cardiomyocytes (*Figure 3f and h–j*), which was comparable to that observed in WT. Similarly, dispersion observed in S2814A cardiomyocytes treated with isoproterenol was reversed by the addition of H89 (*Figure 3g–j*). These observations support that 6MB and H89 treatments exerted their effects on RyR configuration via modulation of PKA activity, without contribution from the CaMKII phosphorylation site on the RyR. Taken together, these data indicate that prolonged PKA-dependent phosphorylation of RyRs is sufficient to drive RyR cluster fragmentation, but that both PKA and CaMKII concertedly drive this process during β-AR stimulation.

## β-Stimulation-induced RyR dispersion alters Ca²⁺ sparks, transients, and waves

We next assessed the functional consequences of RyR dispersion. We examined spontaneous Ca$^{2+}$ sparks since these events occur almost exclusively at dyads (*Louch et al., 2013*), where RyR arrangements appear to be sensitive to β-AR stimulation (*Figure 1f*). Previous work has shown that acute β-AR activation increases the frequency of spontaneous Ca$^{2+}$ sparks, while diverse effects on spark geometry are reported (*Laasmaa et al., 2019*; *Tanaka et al., 1997*; *Viatchenko-Karpinski and Györke,*

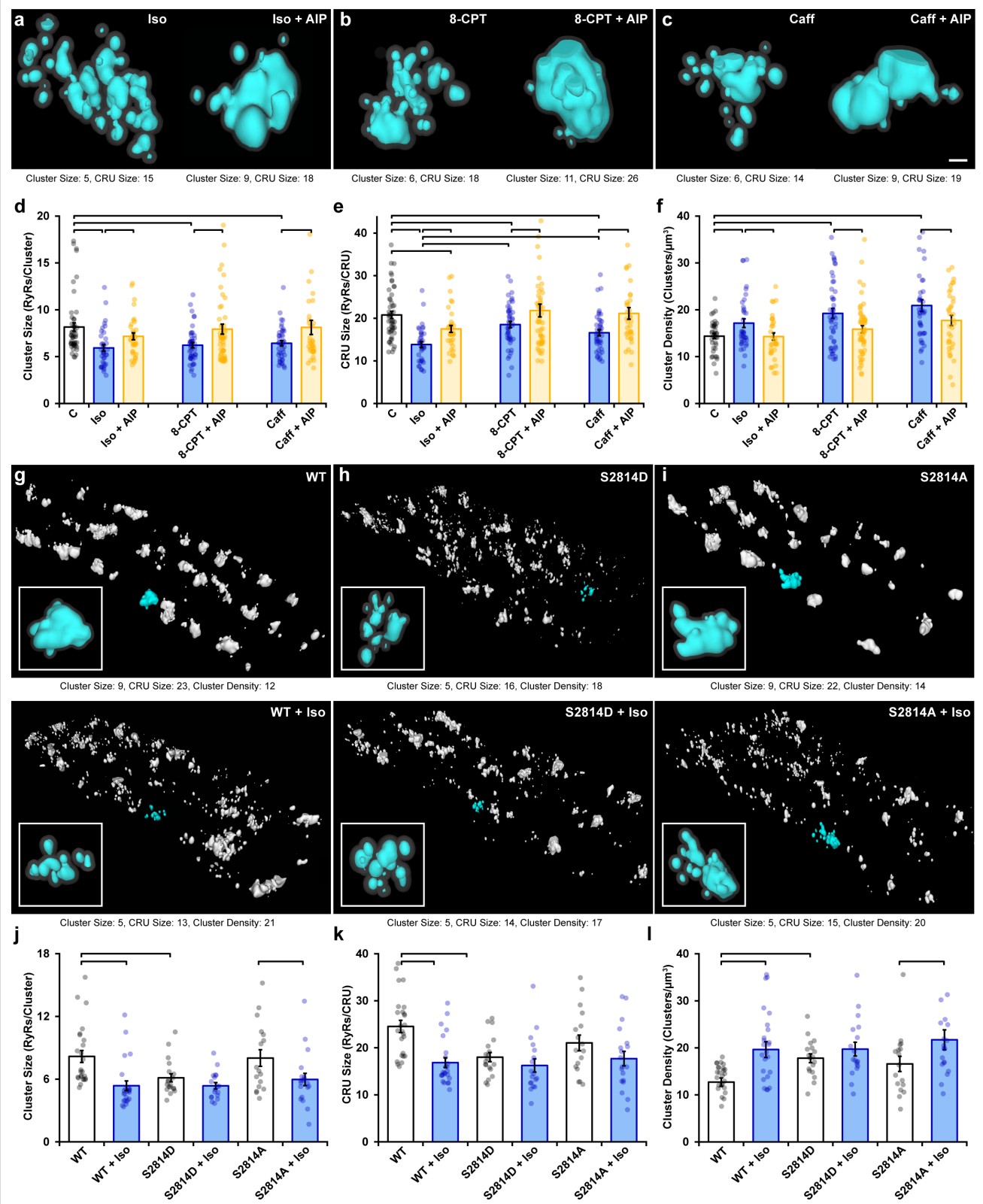

**Figure 2.** Ca²⁺/calmodulin-dependent protein kinase II (CaMKII) activation promotes ryanodine receptor (RyR) cluster dispersion. (**a–c**) Representative Ca²⁺ release units (CRUs) imaged in cardiomyocytes treated with isoproterenol (100 nM), 8-CPT (10 μM), or caffeine (0.5 mM) for 1 hr (left panels) or with inclusion of the CaMKII inhibitor AIP (2 μM) for an additional hour (right panels). Scale bar: 100 nm. (**d– f**) Induced RyR cluster dispersion was reversed by CaMKII inhibition, as indicated by measurements of RyR cluster size, CRU size, and cluster density (control: n_cells = 50, n_hearts = 6; Iso: n_cells =

*Figure 2 continued on next page*

*Figure 2 continued*

37, n$_{hearts}$ = 5; Iso + AIP: n$_{cells}$ = 37, n$_{hearts}$ = 5; 8-CPT: n$_{cells}$ = 48, n$_{hearts}$ = 5; 8-CPT + AIP: n$_{cells}$ = 52, n$_{hearts}$ = 5; caffeine: n$_{cells}$ = 42, n$_{hearts}$ = 4; caffeine + AIP: n$_{cells}$ = 35, n$_{hearts}$ = 3). (**g–i**) Representative images of RyR organization in cardiomyocytes from wildtype (WT) mice and transgenic mice with constitutively activated (S2814D) or genetically ablated (S2814A) phosphorylation at S2814. Clusters were cropped from a region measuring 5 x 9 x 0.6 µm. Images and quantified experimental data (**j–l**) are presented under baseline conditions and following 60 min isoproterenol stimulation (WT: n$_{cells}$ = 25, n$_{hearts}$ = 2; WT + Iso: n$_{cells}$ = 24, n$_{hearts}$ = 2; S2814D: n$_{cells}$ = 19, n$_{hearts}$ = 2; S2814D + Iso: n$_{cells}$ = 18, n$_{hearts}$ = 2; S2814A: n$_{cells}$ = 17, n$_{hearts}$ = 2; S2814A + Iso: n$_{cells}$ = 19, n$_{hearts}$ = 2). The bar charts present mean measurements ± SEM, with superimposed data points representing averaged values from each cardiomyocyte. Statistical significance (p<0.05) between groups is indicated by a comparison bar.

The online version of this article includes the following figure supplement(s) for figure 2:

**Figure supplement 1.** Quantification of ryanodine receptor (RyR) phosphorylation by flow cytometry.

**Figure supplement 2.** Ryanodine receptor (RyR) localization and Ca$^{2+}$ spark characteristics do not differ between 1 hr and 2 hr isoproterenol treatment.

*2001*). We hypothesized, however, that RyR dispersion during prolonged β-AR stimulation would slow spark kinetics and increase Ca$^{2+}$ leak, as released Ca$^{2+}$ propagates between multiple clusters. Representative Ca$^{2+}$ spark recordings from control and isoproterenol-treated cells are presented in *Figure 4a and b*, respectively, with corresponding spark time courses presented in the right panels. We observed that while spark frequency was augmented throughout the isoproterenol treatment period (*Figure 4c*), there was indeed a progressive slowing of both spark rise time (*Figure 4d*) and duration (*Figure 4e*). Increased overall spark geometry (*Figure 4f and g*) during prolonged isoproterenol treatment was linked to progressively augmented Ca$^{2+}$ leak, as calculated based on spark mass and frequency (*Figure 4h*). Importantly, reversal of RyR dispersion by co-incubation of cells with either AIP or H89 after the initial isoproterenol treatment also reversed slowing of spark kinetics and isoproterenol-induced Ca$^{2+}$ leak (*Figure 4d, e and h*). A tendency toward additive effects was noted when AIP and H89 treatments were combined (*Figure 3—figure supplement 1d–f*). In control experiments, a 2 hr treatment period with isoproterenol alone (matching the total treatment time in AIP and H89 experiments) did not alter Ca$^{2+}$ spark properties beyond values observed at the 1 hr time point (*Figure 2—figure supplement 2c and d*). Taken together, these data indicate that, during prolonged β-AR stimulation, RyR dispersion induced by PKA and CaMKII activity is linked to slowing of Ca$^{2+}$ spark kinetics and increased Ca$^{2+}$ leak.

Changes in RyR localization and function during prolonged β-AR stimulation were additionally linked to alterations in cell-wide Ca$^{2+}$ transients (*Figure 4i*). As expected, isoproterenol treatment acutely increased Ca$^{2+}$ transient magnitude; however, this reversed with continued exposure (*Figure 4j*). This pattern was mirrored by changes in SR Ca$^{2+}$ content (*Figure 4—figure supplement 1*), which was also initially increased by heightened SERCA activity during isoproterenol exposure, but then began to decline as RyR Ca$^{2+}$ leak progressively increased. Similar changes were observed in the synchrony of Ca$^{2+}$ release across the cell. Ca$^{2+}$ release tended to be more synchronous at early stages of isoproterenol treatment (*Figure 4l*), in agreement with previous work indicating that increasing SR Ca$^{2+}$ content and RyR sensitization enable more homogeneous Ca$^{2+}$ transients (*Brette et al., 2004*; *Øyehaug et al., 2013*). However, after prolonged β-AR stimulation, RyR dispersal slowed Ca$^{2+}$ spark kinetics, and declining SR content was coupled to a reversal of the synchronization of Ca$^{2+}$ release observed at earlier time points (*Figure 4l*) and a modest slowing of the rising phase of the transient (*Figure 4k*).

The above data are consistent with the notion that RyR dispersion gradually counters the stimulatory effects of β-AR stimulation on CICR, perhaps in a protective manner. To further examine this hypothesis, we investigated the propensity for spontaneous Ca$^{2+}$ waves during isoproterenol treatment (*Figure 4i*). Indeed, while we observed an expected initial increase in Ca$^{2+}$ wave incidence following isoproterenol application, this reversed after 60 min as wave frequency was reduced by ~30% (*Figure 4m*). Thus, sufficient dispersion of RyRs and accompanying weakening of CICR inhibits wave propagation, suggesting a protective role or RyR rearrangement during prolonged β-AR stimulation.

## RyR cluster dispersion lowers CRU excitability in a mathematical model

To support the above correlative data linking RyR arrangement and Ca$^{2+}$ homeostasis, we employed a mathematical reaction–diffusion model of Ca$^{2+}$ release in the CRU. To this end, we constructed spark models employing experimentally observed CRU morphologies. We then ran the computational spark model with these geometries to examine the resulting sparks. We also examined the likelihood of

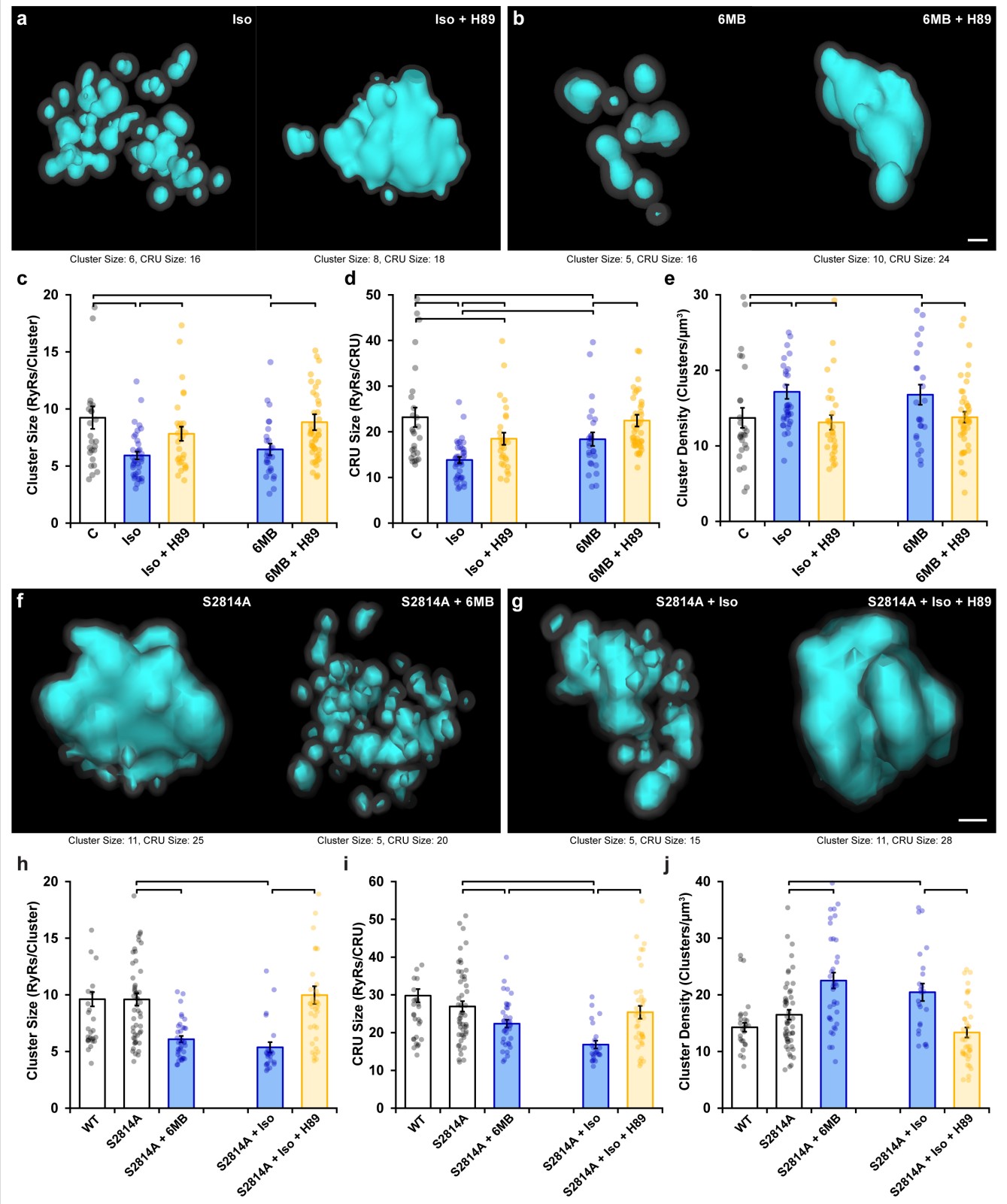

**Figure 3.** Protein kinase A (PKA)-dependent phosphorylation also promotes ryanodine receptor (RyR) cluster fragmentation during β-adrenergic receptor (β-AR) activation. (**a**) Representative Ca$^{2+}$ release units (CRUs) indicate that PKA inhibition (H89, 10 μM) reverses isoproterenol-induced cluster dispersion. (**b**) Direct activation of PKA using 6MB-cAMP (100 μM) can also induce cluster dispersion and is similarly reversed when followed by H89 application. (**c–e**) Data comparing cluster size, CRU size, and cluster density between groups (control: $n_{cells}$ = 26, $n_{hearts}$ = 3; Iso: $n_{cells}$ = 37, $n_{hearts}$ = 5;

*Figure 3 continued on next page*

*Figure 3 continued*

Iso + H89: $n_{cells}$ = 30, $n_{hearts}$ = 4; 6MB: $n_{cells}$ = 26, $n_{hearts}$ = 3; 6MB + H89: $n_{cells}$ = 44, $n_{hearts}$ = 4). (**f**) Representative CRUs indicate that 6MB-cAMP treatment effectively disperses RyR clusters in S2814A cardiomyocytes. (**g**) PKA inhibition also reverses isoproterenol-induced RyR dispersion in S2814A cells. Quantified experimental data are shown in (**h–j**). (WT: $n_{cells}$ = 49, $n_{hearts}$ = 5; S2814A: $n_{cells}$ = 51, $n_{hearts}$ = 5; S2814A + 6MB: $n_{cells}$ = 36, $n_{hearts}$ = 3; S2814A + Iso: $n_{cells}$ = 17, $n_{hearts}$ = 2; S2814A + Iso + H89: $n_{cells}$ = 38, $n_{hearts}$ = 3). The bar charts present mean measurements ± SEM, with superimposed data points representing averaged values from each cardiomyocyte. Statistical significance (p<0.05) between groups is indicated by a comparison bar. Scale bars: 100 nm.

The online version of this article includes the following figure supplement(s) for figure 3:

**Figure supplement 1.** Ryanodine receptor (RyR) cluster organization and $Ca^{2+}$ spark characteristics following AIP, H89, or dual kinase inhibition in normal rat cardiomyocytes.

'undetectable' local release events whose amplitude is too small to be detected in confocal images. While difficult to measure experimentally, this non-spark-mediated leak can importantly affect local SR load.

Using paired images of RyRs (3D dSTORM) and t-tubules (confocal microscopy), we generated four computational CRU geometries that were representative of the range of RyR dispersion observed in cells (*Figure 5a*, *Supplementary file 1*). For each geometry, a series of 400 stochastic spark simulations was performed with and without the regulatory effects of β-AR stimulation on RyR sensitivity and SR load included (*Figure 5b*). Other model parameters are described in *Supplementary file 2* and *Supplementary file 3*.

Each simulated $Ca^{2+}$ release event was classified as being an observable spark if its $\Delta F/F_0$ was ≥ 0.3. However, we also noted the occurrence of smaller modeled release events not observable in experiments. These sub-spark events were characterized as $Ca^{2+}$ quarks (*Brochet et al., 2011*) (0.1 < $\Delta F/F_0$ < 0.3) or failed sparks if their $Ca^{2+}$ release was negligible ($\Delta F/F_0$ ≤ 0.1; *Figure 5b*). We defined spark fidelity as the proportion of $Ca^{2+}$ release events that manifested as full sparks (*Figure 5c*). With baseline model parameters, RyR cluster configuration was observed to have a large impact on spark fidelity. Indeed, the solid CRU arrangement reproducibly generated full sparks, and thus spark fidelity was high (*Figure 5b*, top panels, *Figure 5c*). However, more quarks and failed sparks were observed in the CRUs with minor and moderate RyR dispersion, and in the CRU with major fragmentation, no full $Ca^{2+}$ sparks were generated. Indeed, spark fidelity declined steadily with increased fragmentation (95% CI in solid CRU: 28.6–37.8% vs. majorly fragmented CRU: 0–1.2%, *Figure 5c*). However, β-AR stimulation, as performed in experiments, augments both RyR sensitivity and, particularly at early stages, SR $Ca^{2+}$ content. Increasing channel sensitivity and $Ca^{2+}$ release flux accordingly in the model increased the likelihood that dispersed CRU configurations successfully triggered $Ca^{2+}$ sparks (*Figure 5b*, bottom panels, *Figure 5c*). Thus, inter-RyR communication is improved during β-AR stimulation to enable a greater fraction of RyRs to be activated in fragmented CRUs. These findings support the experimental observation that spark-mediated leak is markedly increased even during prolonged β-AR when CRU configurations are dispersed (*Figure 4h*).

We further employed the model to calculate the proportion of total RyR-mediated $Ca^{2+}$ leak that is 'silent' rather than spark-mediated (*Figure 5d*). Due to declining spark fidelity, we observed that although total $Ca^{2+}$ released from a given CRU decreased with greater fragmentation, the proportion of leak that is silent increased dramatically. When including the effects of β-AR stimulation, the ratio of silent leak was strongly reduced for the mildly to moderately fragmented CRUs. For the majorly fragmented CRU, however, there remained a considerable amount of silent leak (majorly fragmented w/β-AR: 37.5% silent leak, *Figure 5d*) linked to partial activation of the CRU ($Ca^{2+}$ quarks). These results suggest that when CRUs become markedly dispersed during chronic β-AR activation, both spark-mediated and silent leak contribute to declining SR content (*Figure 4—figure supplement 1*) linked to protection against $Ca^{2+}$ waves (*Figure 4m*).

We next examined the characteristics of successful sparks generated by the model to more closely interrogate the relationship between CRU structure and function. Individual simulation results are presented alongside experimental data in *Figure 5—figure supplement 1*. Minor to moderate degrees of RyR dispersion had rather little impact on spark amplitude, and simulated β-AR stimulation similarly increased spark magnitude for these CRUs (*Figure 5e*). Only majorly fragmented CRUs exhibited smaller $Ca^{2+}$ sparks reflecting partial activation of RyRs. Given that experimental data showed a range of dispersed CRU configurations following β-AR stimulation (see example images in *Figure 1*),

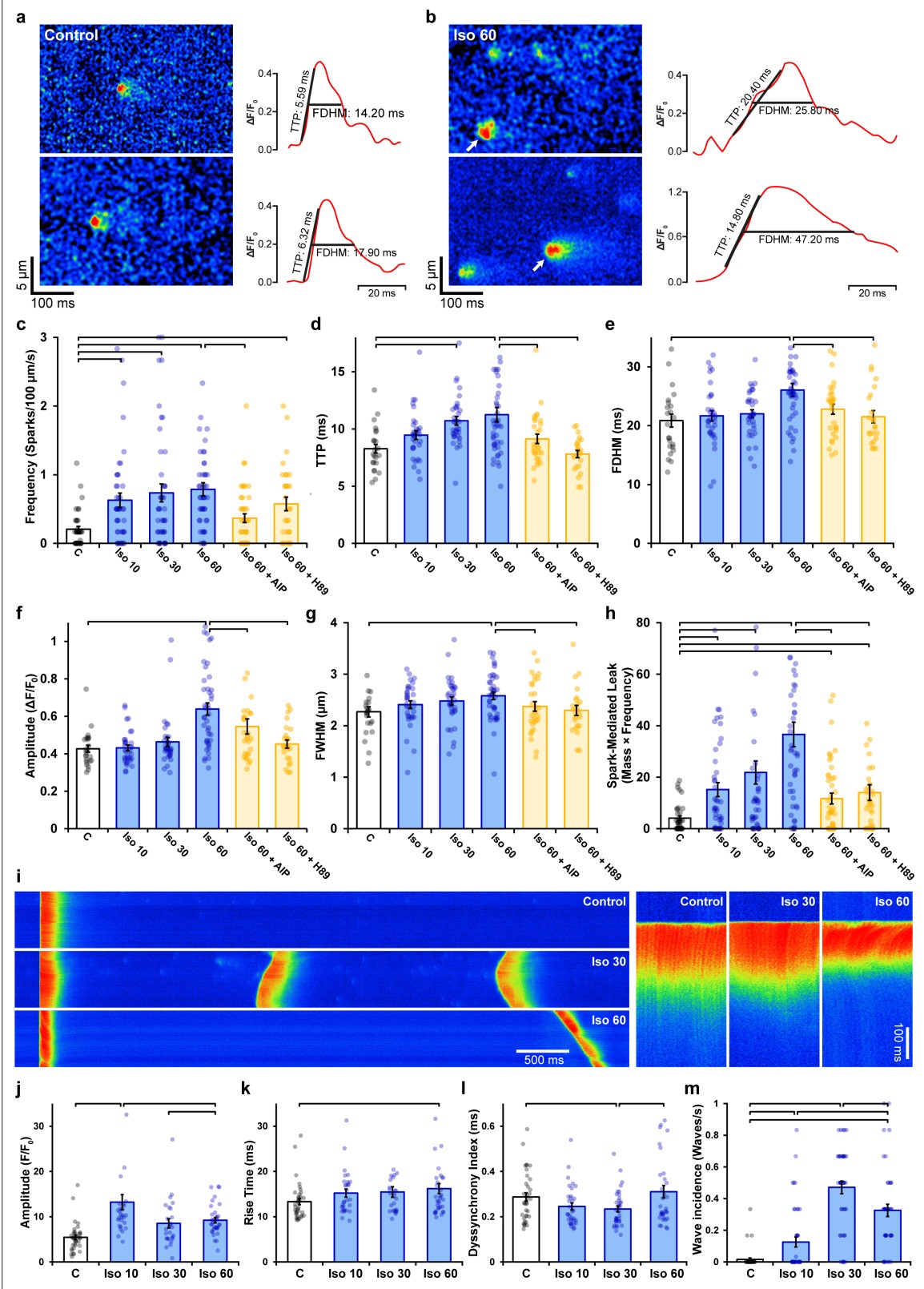

**Figure 4.** Effects of ryanodine receptor (RyR) dispersion on intracellular Ca²⁺ homeostasis are reversed by Ca²⁺/calmodulin-dependent protein kinase II (CaMKII) or protein kinase A (PKA) inhibition. (**a, b**) Representative examples of Ca²⁺ sparks of control and isoproterenol (60 min)-treated cardiomyocytes obtained by confocal line-scan imaging. The time course for each spark is shown on right, with indicated time to peak (TTP) and full duration at half maximum (FDHM) measurements. (**c–h**) Experimental data indicate that increasing spark frequency, geometry, and slowing of kinetics during

*Figure 4 continued on next page*

*Figure 4 continued*

isoproterenol were all reversed by CaMKII or PKA inhibition (control: $n_{sparks}$ = 47, $n_{cells}$ = 38, $n_{hearts}$ = 3; Iso 10: $n_{sparks}$ = 166, $n_{cells}$ = 44, $n_{hearts}$ = 3; Iso 30: $n_{sparks}$ = 194, $n_{cells}$ = 44, $n_{hearts}$ = 3; Iso 60: $n_{sparks}$ = 222, $n_{cells}$ = 47, $n_{hearts}$ = 3; Iso 60 + AIP: $n_{sparks}$ = 147, $n_{cells}$ = 69, $n_{hearts}$ = 5; Iso 60 + H89: $n_{sparks}$ = 135, $n_{cells}$ = 55, $n_{hearts}$ = 4). (**i**) Representative line-scan images illustrating the last in a series of electrically paced $Ca^{2+}$ transients, followed by a pause to examine $Ca^{2+}$ wave generation. Enlargements of the $Ca^{2+}$ transients are shown on right to highlight differences in $Ca^{2+}$ release synchrony. (**j–l**) Data showing initial increases in $Ca^{2+}$ transient amplitude and synchrony were reversed with continued exposure to isoproterenol, while overall release kinetics slowed (control: $n_{cells}$ = 38, $n_{hearts}$ = 3; Iso 10: $n_{cells}$ = 35, $n_{hearts}$ = 3; Iso 30: $n_{cells}$ = 36, $n_{hearts}$ = 3; Iso 60: $n_{cells}$ = 41, $n_{hearts}$ = 3). (**m**) $Ca^{2+}$ wave incidence increased during early time points following isoproterenol treatment, but then reversed (control: $n_{cells}$ = 43, $n_{hearts}$ = 3; Iso 10: $n_{cells}$ = 44, $n_{hearts}$ = 3; Iso 30: $n_{cells}$ = 46, $n_{hearts}$ = 3; Iso 60: $n_{cells}$ = 44, $n_{hearts}$ = 3). The bar charts present mean measurements ± SEM, with superimposed data points representing averaged values from each cardiomyocyte. Statistical significance (p<0.05) between groups is indicated by a comparison bar.

The online version of this article includes the following figure supplement(s) for figure 4:

**Figure supplement 1.** Isoproterenol elicits a biphasic increase in sarcoplasmic reticulum (SR) $Ca^{2+}$ content.

the modeling supports that mean spark amplitude was increased in this condition (***Figure 4f***). The time to peak (TTP) of sparks increased with the degree of dispersion in our model (***Figure 5f***), and, under simulated β-AR activation, this was primarily due to the sensitized RyRs' improved ability to sustain regenerative release between clusters. Indeed, in simulations where SR load was systematically altered with constant RyR $Ca^{2+}$ sensitivity, spark kinetics were not markedly affected (***Figure 5—figure supplement 2***). These findings support that RyR cluster dispersion during prolonged β-AR activation is directly associated with larger and slower $Ca^{2+}$ sparks observed experimentally.

Taken together, the modeling results critically link changes in CRU configuration and spark morphology; phenomena that were observed separately in experiments. However, the modeling data also support the contention from experimental data that RyR cluster fragmentation can reduce arrhythmogenic spontaneous $Ca^{2+}$ release ($Ca^{2+}$ waves). Here, the antiarrhythmic actions of RyR dispersion appear to be linked to a reduction in SR content resulting from an increase in both spark-mediated and silent leak. We postulate that local regions of the cell with dispersed CRUs and lowered SR content may act as a physical barrier or 'fire break' to impede $Ca^{2+}$ wave propagation.

## CaMKII and PKA activation promote RyR dispersion and $Ca^{2+}$ dysregulation during HF

We next examined whether dispersion of RyRs reported during HF (***Kolstad et al., 2018***) similarly results from RyR hyperphosphorylation. Previous work has indeed linked CaMKII (***Ling et al., 2009***; ***Zhang et al., 2003***) and PKA activity (***Marx et al., 2000***) to RyR dysfunction in this disease. We first examined RyR phosphorylation status in rats with post-infarction HF compared to sham-operated controls (see ***Supplementary file 4*** for animal characteristics). Western blotting revealed no significant change in total RyR expression and only a tendency toward increased PKA phosphorylation at S2808 (***Figure 6a and b***, ***Figure 6—source data 1***). However, phosphorylation at S2814 – a site linked to CaMKII activation – was significantly increased by ~55% in HF (***Figure 6—source data 1***).

3D dSTORM imaging of Sham and HF cardiomyocytes (***Figure 6c***) revealed significant RyR dispersion in failing cells, manifested at both the cluster level (***Figure 6d***; sham: 7.93 ± 0.46 vs. HF: 5.16 ± 0.27 RyRs/cluster) and CRU level (***Figure 6e***; sham: 20.79 ± 1.09 vs. HF: 14.82 ± 1.01 RyRs/CRU). These structural changes are consistent with a previous 2D super-resolution study that utilized the same pathological model (***Kolstad et al., 2018***). Next, we subjected cells to either CaMKII or PKA inhibition to assess whether disrupted RyR organization could be reversed. Indeed, inhibiting CaMKII phosphorylation with AIP in HF cells restored RyR organization to sham values, as evidenced by a 49% increase in cluster size and a 47% increase in CRU size (***Figure 6d and e***). To a lesser extent, the addition of H89 to failing cells also significantly reversed RyR dispersion at the cluster level (28% increase) and CRU level (19% increase). Correlative imaging of RyRs and t-tubules (***Figure 6c***) revealed a significantly smaller proportion of dyadic clusters in HF compared to sham (***Figure 6f***); an expected finding since reduced t-tubule density in this model results in the formation of 'orphaned' RyRs (***Frisk et al., 2016***; ***Song et al., 2006***). The smaller RyR clusters observed in failing cells were found to exclusively occur at dyadic sites (sham: 9.09 ± 0.47 vs. HF: 6.81 ± 0.44 RyRs/cluster), where dispersed RyR arrangements were again reversible by CaMKII or PKA inhibition (***Figure 6g***). The striking similarity of these findings to experiments employing prolonged isoproterenol exposure (***Figures 1–4***) supports that fragmentation of RyR clusters during HF results from increased activation of both CaMKII and

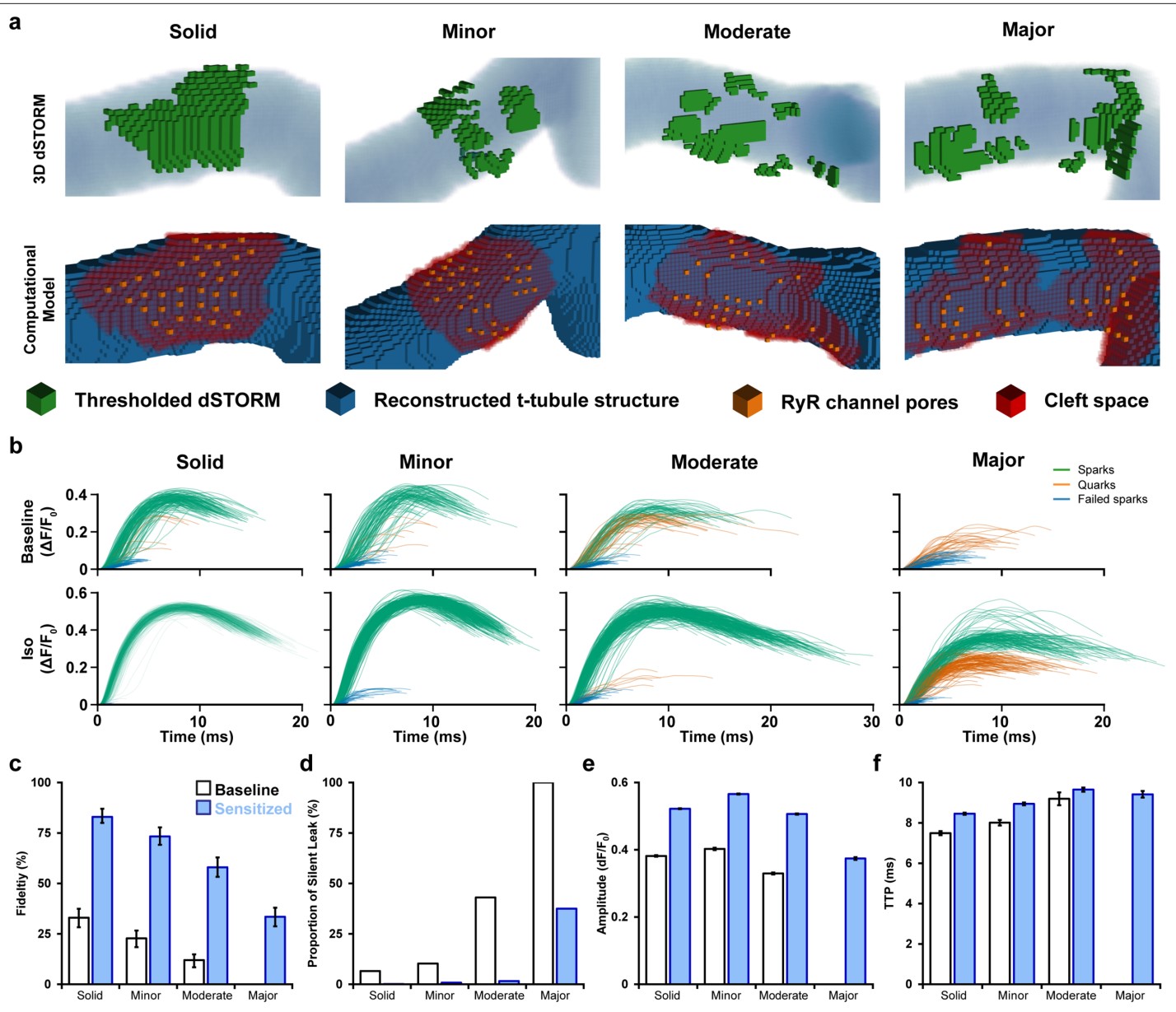

**Figure 5.** Mathematical modeling linking changes in ryanodine receptor (RyR) configuration and function during β-adrenergic stimulation. (**a**) 3D computational volumetric meshes built from correlative imaging of the t-tubular network (confocal microscopy) and RyRs (3D dSTORM). Constructed geometries consisted of 10 × 10 × 10 nm voxels and a 1 × 1 × 1 μm volume. The four geometries were chosen to represent the range of cluster dispersion seen in imaging (see **Supplementary file 1** for details). (Top row) Reconstructed t-tubular structures (blue) are illustrated with superimposed thresholded RyR signals (green). (Bottom row) Corresponding computational geometries, with indicated RyR channel pores (orange) and cleft space (red) defined by morphological dilation around the RyRs. The four geometries were selected to represent the range of RyR dispersion observed in experiments, ranging from solid to majorly fragmented Ca²⁺ release units (CRUs) (see 'Materials and methods'). (**b**) In total, 400 stochastic spark simulations were performed in each geometry with baseline model parameters, and another 400 were conducted with the regulatory effects of β-adrenergic receptor (β-AR) stimulation added (RyR sensitization and increased sarcoplasmic reticulum [SR] content). The $\Delta F/F_0$ time course is shown for each spark. The maximal amplitude used to define the release as an experimentally observable spark event ($\Delta F/F_0 \geq 0.3$), a sub-spark quark event ($\Delta F/F_0 \geq 0.1$), or a failed spark ($\Delta F/F_0 < 0.1$). (**c, d**) The classification of release into spark and non-spark events was used to estimate spark fidelity and the ratio of leak that is 'silent' (i.e., sub-spark release) in each simulation case. (**e, f**) Average amplitude and time to peak (TTP) measurements for observable sparks. Unshaded and shaded bars in (**c–f**) denote baseline and sensitized (simulated β-AR activation) RyR conditions, respectively. Error bars in (**e**) and (**f**) show SEM, while the bars in (**c**) indicate the 95% Agresti–Coull confidence interval.

The online version of this article includes the following figure supplement(s) for figure 5:

**Figure supplement 1.** Comparison of experimentally measured and simulated Ca²⁺ sparks.

**Figure supplement 2.** Model behavior at different sarcoplasmic reticulum (SR) loads.

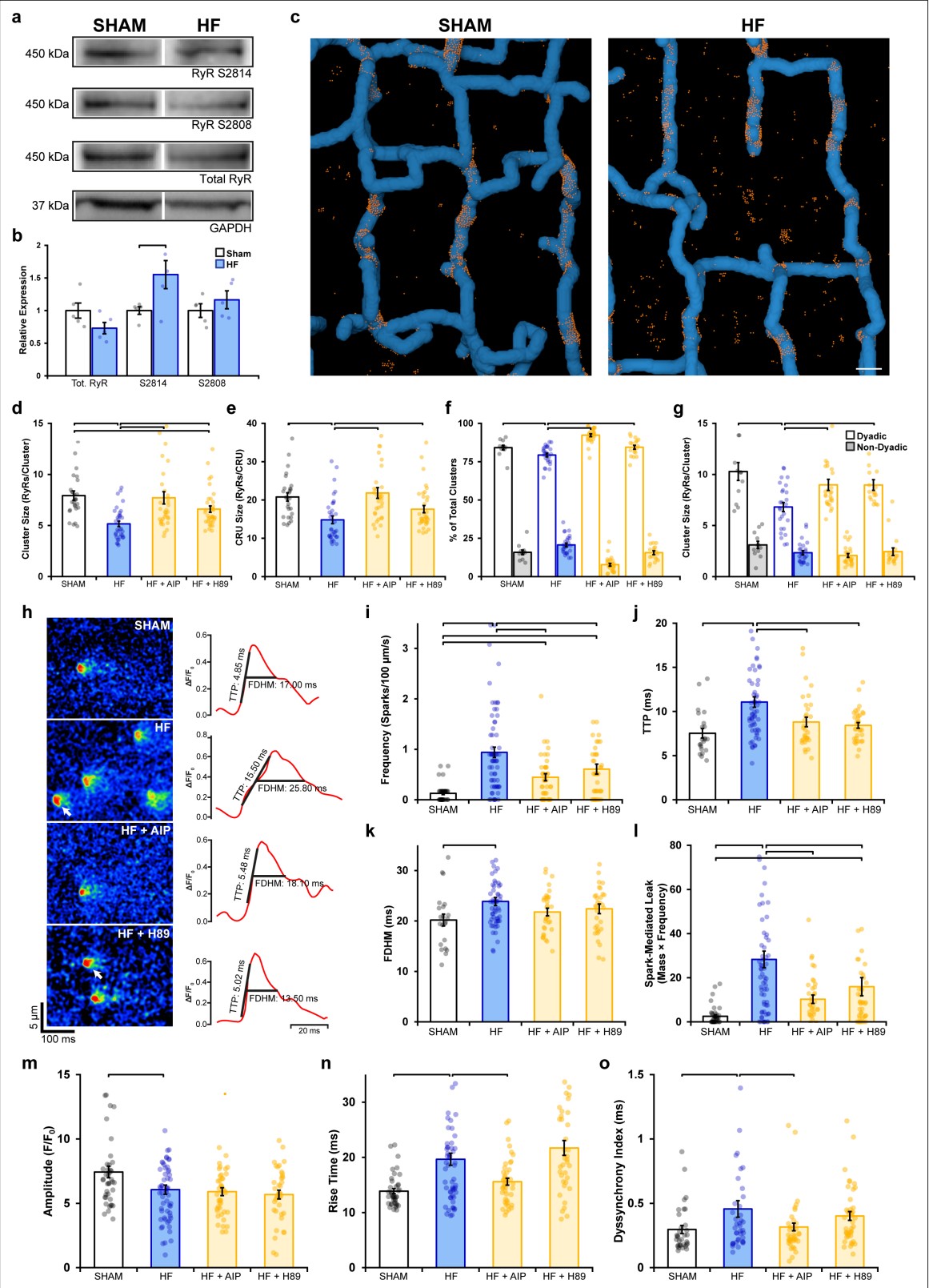

**Figure 6.** Remodeling of ryanodine receptor (RyR) organization and function during heart failure (HF) are reversed by inhibiting phosphorylation. (**a, b**) Representative Western blots and quantification of the relative expression of $Ca^{2+}$/calmodulin-dependent protein kinase II (CaMKII)-phosphorylated RyR (S2814), protein kinase A (PKA)-phosphorylated RyR (S2808), and total RyR in sham and HF cardiomyocytes (sham: $n_{hearts}$ = 5; HF: $n_{hearts}$ = 5). (**c**) 3D reconstruction of dyads in sham and failing cells based on correlative imaging of t-tubules (blue, confocal microscopy) and RyRs (orange, 3D dSTORM).

*Figure 6 continued on next page*

*Figure 6 continued*

Scale bar: 500 nm. (**d, e**) RyR cluster and Ca²⁺ release unit (CRU) size measurements in cardiomyocytes from sham and HF cells, at baseline and following inhibition of CaMKII (AIP) or PKA (H89) (sham: $n_{cells}$ = 27, $n_{hearts}$ = 4; HF: $n_{cells}$ = 33, $n_{hearts}$ = 4; HF + AIP: $n_{cells}$ = 33, $n_{hearts}$ = 4; HF + H89: $n_{cells}$ = 35, $n_{hearts}$ = 4). Dyadic/non-dyadic cluster proportion (**f**) and sizes (**g**) in sham and HF cells, at baseline and following CaMKII or PKA blockade. (**h**) Representative Ca²⁺ spark examples from sham and HF cardiomyocytes with or without CaMKII inhibition (AIP) or PKA inhibition (H89). The time course of each spark is shown on right. (**i–l**) Measurements of spark frequency, time to peak (TTP), duration, and spark-mediated leak in sham and HF cells at baseline and following treatment with AIP or PKA (sham: $n_{sparks}$ = 32, $n_{cells}$ = 42, $n_{hearts}$ = 3; HF: $n_{sparks}$ = 476, $n_{cells}$ = 65, $n_{hearts}$ = 5; HF + AIP: $n_{sparks}$ = 163, $n_{cells}$ = 63, $n_{hearts}$ = 5; HF + H89: $n_{sparks}$ = 196, $n_{cells}$ = 43, $n_{hearts}$ = 3). (**m–o**) Measurements of cell-wide Ca²⁺ transient magnitude, TTP (rise time), and dyssynchrony in sham and HF cells in the presence and absence of AIP or H89 (sham: $n_{cells}$ = 41, $n_{hearts}$ = 4; HF: $n_{cells}$ = 59, $n_{hearts}$ = 4; HF + AIP: $n_{cells}$ = 43, $n_{hearts}$ = 4; HF + H89: $n_{cells}$ = 38, $n_{hearts}$ = 3). The bar charts present mean measurements ± SEM. For Western blotting data, superimposed data points represent measurements from individual hearts (**b**). For all graphs, data points represent averaged values from each cardiomyocyte. Statistical significance (p<0.05) between groups is indicated by a comparison bar.

The online version of this article includes the following source data and figure supplement(s) for figure 6:

**Source data 1.** Raw Western blots of total RyR, Ser2814 and Ser2808 expression.

**Figure supplement 1.** Ca²⁺ transients in sham and heart failure (HF) myocytes.

PKA, likely as a consequence of the chronic β-AR stimulation that is well established in this condition (*Ganguly et al., 1997*).

Finally, we examined whether alterations in RyR Ca²⁺ release during HF could be linked to RyR phosphorylation and dispersion. Representative recordings of Ca²⁺ sparks are presented in *Figure 6h*, together with temporal profiles shown on right. Failing cardiomyocytes exhibited significantly increased frequency (*Figure 6i*), slower rise time (*Figure 6j*), and longer duration sparks (*Figure 6k*) than in sham cells, which summated to marked augmentation of spark-mediated leak (*Figure 6l*). All of these changes were reversed by inhibiting CaMKII or PKA, although the effects of CaMKII inhibition were the most marked (*Figure 6h–l*). Overall Ca²⁺ transients were of lower magnitude in failing cardiomyocytes (*Figure 6m*), and Ca²⁺ release was observed to be slowed and dyssynchronous (*Figure 6n and o*, see *Figure 6—figure supplement 1* for representative examples). CaMKII inhibition again reversed changes in Ca²⁺ transient kinetics and synchrony in failing cells (*Figure 6n and o*). Taken together, these findings identify phosphorylation-dependent RyR dispersion as a key driver of RyR dysfunction in HF and particularly implicate CaMKII overactivity in this process.

## Discussion

In this work, we identify an important new mechanism by which RyR organization and function are controlled by β-AR stimulation. Specifically, we show that during prolonged β-AR activation, as occurs during HF, there is a progressive dispersion of RyR clusters. This fragmentation is driven by both CaMKII- and PKA-dependent phosphorylation of the channel, and is reversible by inhibiting these kinases. Functionally, our results indicate that dispersed CRUs exhibit increased 'silent' Ca²⁺ leak and low spark fidelity. When Ca²⁺ sparks are successfully generated by dispersed CRUs, they exhibit slowed kinetics and reduced magnitude. These changes are in turn coupled with smaller and desynchronized Ca²⁺ transients, which are hallmarks of failing cells. However, we also observed protective actions of CRU dispersion as lowered spark fidelity and declining SR Ca²⁺ content protected against Ca²⁺ wave generation. Thus, RyR dispersal appears to be aimed at curbing arrhythmogenesis during prolonged β-AR stimulation, but comes at the expense of systolic function, and this is particularly detrimental in the setting of HF.

While the precise nature of inter-RyR dynamics remains incompletely understood, several previous studies have suggested that there are important structural determinants of CRU function (*Galice et al., 2018*; *Kolstad et al., 2018*; *Walker et al., 2014*; *Zima et al., 2008*). Of note, recent work by *Galice et al., 2018* measured RyR cluster size and Ca²⁺ sparks simultaneously, finding that spark frequency depends steeply on cluster size, while spark amplitudes appear to be independent of size above a certain threshold. The authors argued that RyR cluster sizes must strike a balance between being sufficiently large to ensure high-fidelity CICR, but not so large as to promote arrhythmogenic behavior. When RyR channels are sensitized or Ca²⁺ levels are elevated, this balance shifts, increasing the propensity for sparks, diastolic leak, and waves. Extending from these findings, our present

experimental and mathematical modeling results suggest that RyR cluster dispersion compensates for this shift by physically inhibiting $Ca^{2+}$ wave propagation, but to the detriment of spark fidelity.

Our data show that marked RyR cluster dispersion occurs only after prolonged β-AR stimulation, which likely explains why previous, acute treatment studies have not detected similar changes in RyR arrangement or function. For example, *Asghari et al., 2020* recently performed detailed analysis of internal RyR positioning, but within only 10 min of channel phosphorylation. Nevertheless, these authors did observe changes in the precise positioning of neighboring RyRs within this time frame as a checkerboard arrangement of RyRs was favored over side-by-side arrangements. Both arrangements involve SRPY and P1 domain–domain interactions between adjacent receptors (*Cabra et al., 2016*). A shift in the configuration of these domains, for example, during phosphorylation, might be envisioned to reduce the stability of RyR dimerization, that is, reducing RyR 'stickiness.' Indeed, *Asghari et al., 2020* observed a modest rightward shift in the RyR–RyR nearest-neighbor distance early after initiation of phosphorylation, which could represent the earliest stages of interior cluster breakup that we have observed at later time points. However, neither we (*Figure 1—figure supplement 2*) nor Asghari et al. observed similar rearrangement of RyRs at the cell surface during β-AR stimulation. Rather both groups observed a tendency toward *increased* cluster size at the cell surface, starting from early time points after initiation of β-AR stimulation. The reason for the disparate response to phosphorylation between surface and interior clusters remains unclear. However, there may be unique features of RyRs and/or their partner proteins that render their positioning phosphorylation-sensitive. FKBP12.6 is one such candidate as it has been shown to be directly involved in domain–domain interactions between adjacent RyRs (*Cabra et al., 2016*) and to fine-tune RyR positions in directions that are opposite those of phosphorylation (*Asghari et al., 2020*). Another possible explanation is the cellular distribution of different subtypes of β-AR. Previously, it has been shown that while $β_1$-AR is ubiquitously distributed across the cardiomyocyte surface, $β_2$-AR are only present in the t-tubules (*Nikolaev et al., 2010*). Thus, it is possible that it is solely $β_2$-AR activation that is responsible for signaling RyR cluster dispersion; a hypothesis that will be investigated in future work.

It is well established that prolonged β-AR stimulation is linked to time-dependent desensitization of the signaling pathway; a process that involves G protein-coupled receptor kinases and β-arrestins that promote internalization and downregulation of the receptors (*Kayki-Mutlu and Koch, 2021*). Our results indicate that this process was likely initiated in our experiments since RyR phosphorylation at ser-2814 and ser-2808 both exhibited bell-shaped responses during the 1 hr treatment protocol (*Figure 2—figure supplement 1*). A similar response was observed in measurements of SERCA activity (*Figure 4—figure supplement 1c*). Importantly, RyR phosphorylation by CaMKII and/or PKA remained elevated above control levels at the end of the protocol. These heightened phosphorylation levels apparently remain sufficient to support RyR dispersion since inhibition of either kinase following β-AR stimulation caused RyRs to re-cluster (*Figures 2a–f and 3a-e*). The regulatory role of CaMKII and PKA on RyR dispersion was additionally corroborated by experiments showing that selective activation of either kinase induced fragmented CRU arrangements. Furthermore, transgenic mice with constitutively active S2814 were found to have significantly dispersed clusters at baseline compared to their WT counterparts (*Figure 2g–l*). Notably, genetic ablation of S2814 did not appear to inhibit the effects of β-AR stimulation, which continued to reduce RyR cluster size and increase cluster density. Taken together, these findings point to a more complicated mechanism responsible for cluster dispersion than direct phosphorylation of the RyR at the S2814 and S2808 residues by CaMKII and PKA, respectively. Rather, it seems likely that CaMKII and PKA act interdependently on the RyR. Other studies have also suggested cross-talk between the two sites, with one observing changes to S2814 phosphorylation in an S2808 mutant (*Wang et al., 2015*) and another demonstrating increased PKA activity at S2808 following S2814 phosphorylation (*Haji-Ghassemi et al., 2019*). In HF cells, we detected an increase in RyR phosphorylation at S2814, but only a minor trend towards higher phosphorylation at S2808. Still, inhibition of either CaMKII or PKA reversed cluster dispersion in these cells, although the extent of reversion following AIP administration was more complete than that obtained with H89. These results appear to be consistent with studies suggesting that PKA may not be the principal determinant of RyR hyperphosphorylation in HF (*Xiao et al., 2005*; *Zhang et al., 2003*), but also indicate that CaMKII and PKA control RyR clustering in a coordinated manner.

Although we observed a close relationship between RyR organization and function in our experiments, we relied on mathematical modeling to directly interrogate the functional implications of

cluster dispersion vs. other regulatory effects of β-AR stimulation. The modeling results indicated that RyR cluster fragmentation is sufficient to significantly weaken intra-CRU RyR communication. Notably, these structural-driven changes in function were opposite those induced by regulatory changes as increased RyR sensitivity to cytosolic Ca$^{2+}$ and increased SR Ca$^{2+}$ content increased intra-CRU RyR communication. These observations are consistent with the view that while acute β-AR stimulation effectively augments CICR, progressive RyR dispersion and accompanying decline of SR Ca$^{2+}$ content during continued exposure serves to gradually reverse these changes as Ca$^{2+}$ release fidelity, magnitude, and kinetics are compromised (*Figure 5c–f*). In healthy cardiomyocytes, this functional decline was not overly severe as Ca$^{2+}$ transients measured experimentally remained somewhat larger than in untreated cells, and Ca$^{2+}$ release kinetics were only modestly slowed (*Figure 4j and k*). However, during HF, more detrimental consequences of RyR dispersion are expected as SR content is reduced below normal levels. Thus, compensatory changes in RyR function observed in healthy cells during β-AR stimulation are expected to be less present and impairment of CICR more robust. Future work should be aimed at directly linking β-adrenergic tone, RyR dispersion, and Ca$^{2+}$ signaling impairments in experimental HF and establish the reversibility of this relationship by the application of β-blockers.

Experimental studies have frequently linked increases in spark frequency to higher incidence of Ca$^{2+}$ waves (*Fernández-Velasco et al., 2009*; *Terentyev et al., 2006*). Here, we also report a marked increase in both spark and wave incidence during the early stages of β-AR stimulation (*Figure 4i and m*). However, while both spark frequency and mass continued to increase with prolonged treatment, wave incidence significantly declined. While this finding may at first seem counterintuitive, a similar 'paradoxical' observation was reported in a canine model of left ventricular hypertrophy (*Song et al., 2005*). The authors hypothesized that disconnects between junctional SR clusters and LTCCs could lead to spatial heterogeneity in SR Ca$^{2+}$ content, enabling regions with lower SR Ca$^{2+}$ to act as 'fire breaks' against wave propagation. We suggest that an analogous mechanism explains our findings, but which may include both reduced SR content and dispersed CRUs acting as a physical barrier to Ca$^{2+}$ wave propagation. Since accumulating evidence supports that RyR positioning is tightly regulated, we believe that RyR dispersion in healthy cardiomyocytes is, by design, aimed at curbing wave generation during prolonged β-AR stimulation. This is an important consideration if we are to consider targeting RyR localization in disease. While increasing RyR–RyR 'stickiness' in failing cells would be expected to augment RyR clusters sizes, inter-channel collaboration, and systolic function, our data suggest that such effects may also increase susceptibility to arrhythmia.

In summary, we observed that RyR clusters progressively disperse during protracted β-AR activation due to channel phosphorylation by both CaMKII and PKA. Experimental and mathematical modeling results linked this dyadic rearrangement to declining efficacy of CICR, including slowed and reduced magnitude Ca$^{2+}$ release, but also protection against pro-arrhythmic Ca$^{2+}$ waves. These findings have important implications for HF and other conditions such as catecholaminergic polymorphic ventricular tachycardia, which are associated with increased phosphorylation of RyRs and risk of arrhythmia.

## Materials and methods
### Transgenic mice
We examined the effects of the CaMKII phosphorylation site S2814 on RyR configuration by employing knock-in transgenic mouse models, where the site was rendered constitutively active (RyR2-S2814D; *van Oort et al., 2010*) or genetically ablated (RyR2-S2814A; *Chelu et al., 2010*). Cells isolated from homozygous mutant mice were used for this study, with comparison made to their respective WT littermates that served as controls.

### Rat model of post-myocardial infarction congestive HF
Left coronary artery ligation was performed to induce large anterolateral myocardial infarctions in male Wistar rats (*Lunde et al., 2012*). Development of HF was verified 6 weeks later using a Vevo 2100 echocardiography imaging system (VisualSonics, Toronto, Canada). HF animals were selected based on established criteria (*Sjaastad et al., 2000*), including dilation of the left atrium (diameter > 5 mm) and infarction size above 40% (*Supplementary file 4*). Sham-operated rats served as controls. Sample sizes were determined by power analysis, assuming that only 50% of post-infarction animals

would be included in the final data set, and based on a pilot project of variability in CRU morphology in healthy controls.

## Rat ventricular cardiomyocyte isolation

Isolation of cardiomyocytes was based on the protocol previously described by *Hodne et al., 2017*. Briefly, animals were anesthetized with isoflurane and sacrificed, and hearts were rapidly excised. Each heart was then cannulated and mounted on a constant-flow (3 mL/min) Langendorff setup, perfused with $Ca^{2+}$-free oxygenated solution (in mmol/L: 140 NaCl, 5.4 KCl, 0.5 $MgCl_2$, 0.4 $NaH_2PO_4$, 5 HEPES, 5.5 glucose pH 7.4). Once the heart had been cleared of blood, perfusion was switched to the same solution containing collagenase type II (1.8 mg/mL, Worthington Biochemical Corporation) for 10–12 min at 37°C. Following digestion, left ventricular tissue was dissected and finely cut into 3–4 $mm^3$ pieces. A secondary digestion was performed to liberate additional cells from tissue by transferring approximately 8 mL of tissue and collagenase solution to a 10 mL Falcon tube containing 0.2 mg DNase (LS002006, Worthington) in 500 µL bovine serum albumin (BSA). Cells were subsequently filtered and allowed to pellet in 0.2 mmol/L $Ca^{2+}$.

## Immunofluorescence labeling

Isolated cardiomyocytes were washed with Dulbecco's phosphate-buffered saline (PBS) (no. 4387, BioWhittaker), fixed with 4% paraformaldehyde for 10 min, quenched with 100 µmol/L glycine for 10 min, and permeabilized with 1% Triton X-100 for 10 min. The cells were then plated on glass-bottom dishes (no. 1.5, Ø 14 mm, γ-irradiated, Martek Corporation) that had been coated with laminin (mouse, BD Biosciences) and blocked using Image-iT FX Signal Enhancer (Thermo Fisher Scientific) prior to immunolabeling.

For RyR visualization, cells were incubated with mouse anti-RyR (1:100, MA3-916, Thermo Fisher Scientific) overnight at 4°C. For t-tubule imaging, cells were incubated overnight at 4°C in a mixture of rabbit anti-Cav-3 (1:100, ab2912, Abcam) antibody and a custom rabbit anti-NCX1 antibody as previously described (*Shen et al., 2019*) (1:100, GenScript Corporation, Piscataway, NJ). Secondary antibody labeling was carried out using donkey anti-mouse Alexa Fluor 647 (1:200, A-21237, Thermo Fisher Scientific) and goat anti-rabbit Alexa Fluor 488 (1:200, A-11070, Thermo Fisher Scientific) antibodies for 1 hr at room temperature. Both primary and secondary antibodies were diluted in a blocking buffer consisting of 2% goat serum and 0.02% $NaN_3$ in PBS.

## 3D dSTORM super-resolution imaging and reconstruction of RyRs

Alexa Fluor 647-labeled rat ventricular cardiomyocytes were submersed in an imaging buffer containing 20% VectaShield (H-1000, Vector Laboratories) diluted in Tris-glycerol (5% v/v TRIS 1 M pH 8 in glycerol, Sigma-Aldrich). This composition has been previously shown to produce comparable, if not superior, quality of dSTORM images compared with conventional oxygen scavenging-dependent systems (*Olivier et al., 2013*).

Cardiomyocytes were imaged using the Zeiss ELYRA/LSM 710 system (Carl Zeiss, Jena, Germany). A diode laser (150 mW, 642 nm) illuminated the sample via a plan-apo 63 × 1.4 NA oil objective, configured in a highly inclined and laminated optical sheet (HiLo). 3D imaging was achieved utilizing Phase Ramp Imaging Localization Microscopy (PRILM) technology (*Baddeley et al., 2011*). Fluorescence emission > 655 nm was collected with an iXon 897 back-thinned EMCCD camera (Andor Technology, Belfast). A sequence of 15,000 frames was acquired for each cell at a frame exposure time of 40 ms. Throughout image acquisition, a piezo-operated Definite Focus system was employed to autocorrect for axial drift.

Reconstruction of dSTORM data was carried out using the 'PALM Processing' module in the ZEN Black software (Zeiss). In short, an experimental 3D PSF with an axial range of 4 µm was acquired using 100 nm TetraSpecks (T7279, Thermo Fisher Scientific). Individual single-molecule events were detected by employing an 11 pixel circular mask, with a signal-to-background noise ratio of 6. Drift correction was performed using a five-segment piecewise linear function, and a text-based points table was then generated containing the x, y, and z coordinates of each localization event. In order to minimize the inclusion of clusters with larger localization error, events from only the central 600 nm of the 4 µm stack were included (*Baddeley et al., 2011*). Lastly, the points table was processed via a custom-written Python script to generate a pixel-based image whereby individual events were

represented with a Gaussian function centered at the event coordinates and a width corresponding to its lateral and axial precision values (in nm). The resulting intensity data stack was then thresholded using the Ostsu method and output as a 600 nm z-stack with a voxel size of 30 nm, so that each voxel of the resulting 3D binary mask stack contained no more than a single RyR.

## Quantitative analysis of 3D RyR cluster characteristics

3D quantification of RyR organization was performed as previously described (*Shen et al., 2019*). In short, by combining PRILM dSTORM and fluorescent event registration, we were able to discern the irregular shapes of interior RyR arrangements and estimate RyR cluster and CRU sizes. Based on mathematical modeling by *Sobie et al., 2006*, RyR clusters with edges localized within 100 nm were assigned to the same CRU. RyR cluster density was calculated and normalized by totaling the number of clusters within a 2 × 2 × 0.6 μm volume positioned within the cell interior. The volumetric density of RyRs was determined on a per-cell basis by multiplying the average cluster size by the cluster density.

## Correlative imaging of RyRs and t-tubules for dyad reconstruction

The Zeiss ELYRA system was also used for confocal imaging of RyRs (Alexa Fluor 647, 633 nm laser) and t-tubules (Alexa Fluor 488, 488 nm laser) for 3D correlative reconstruction of dyads. Prior to dSTORM imaging of RyRs, a 3-μm-thick confocal stack centered on the same region was acquired (frame size: 1024 × 1024 pixels; pixel size: 50 nm; z-spacing: 200 nm per slice). Deconvolution of confocal images was performed using Huygens Essential software (SVI, the Netherlands), with a signal-to-noise ratio of 5. Using the confocally imaged RyR channel as a reference, confocally measured t-tubule distributions were correlated to dSTORM-derived RyR positions. This was done with a custom-written Python script that corrected for both lateral, axial, translational, and scaling differences between the two imaging modalities to optimally align the dSTORM-derived RyR data with the confocal RyR data (*Figure 1—figure supplement 3*).

## 3D geometric rendering of RyRs and t-tubules

To examine the arrangement of RyR clusters in relation to t-tubules, 3D volumetric geometries were constructed from the correlated images of these structures using the same approach as described in our earlier work (*Shen et al., 2019*). Geometries were built using a custom Python script relying on NumPy 1.18 (*van der Walt et al., 2011*), SciPy 1.5 (*Virtanen et al., 2020*), and scikit-image 0.15 (*van der Walt et al., 2014*) for morphological manipulation.

A 3D reconstruction of the t-tubule network was created from confocal NCX1/Cav-3 imaging by first fitting the data to 10 nm × 10 nm × 10 nm voxels, binarizing using an adaptive Gaussian threshold, and then pruning small non-contiguous components (<0.03 μm³). The resulting thresholded structures were skeletonized using a 3D skeletonization algorithm (*Lee et al., 1994*), and re-dilated to form 250-nm-wide cylindrical t-tubules (*Soeller and Cannell, 1999*). The correlated dSTORM imaging data of RyRs were then fitted to the same voxel grid and Otsu thresholded. The majority of the thresholded clusters directly overlapped with the defined dyadic RyR face. Due to inaccuracies in the t-tubule reconstruction as well as limited precision in the underlying confocal imaging, all clusters lying within 150 nm of a reconstructed t-tubule were estimated to be dyadic clusters. More distally localized RyR clusters were deemed to be non-dyadic.

To render the 3D geometric reconstructions and produce geometric meshes for use in mathematical modeling, specific locations of individual RyRs were assigned by projecting the thresholded dyadic cluster data onto a cylindrical shell surrounding the t-tubules, leaving a 1-voxel-wide (10 nm) dyadic cleft. These overlap interfaces were then randomly filled with RyRs such that no channels were closer than a center-to-center distance of 30 nm. As shown in our earlier work, this approach yields RyR numbers that are in good agreement with our experimentally obtained estimates (*Shen et al., 2019*).

To visualize the reconstructed geometry of t-tubules and RyRs, iso-surfaces were generated from the full geometries using the Lewiner Marching-Cubes algorithm (*Lewiner et al., 2003*), smoothed using GAMer 2.0 (*Lee et al., 2020*), and finally rendered using Blender (Blender Foundation, the Netherlands).

For use in computational modeling, an SR network was heuristically added to the voxel-based reconstruction of t-tubules and RyRs. The junctional SR (jSR) terminals for each cluster were defined

by iterative morphological binary dilation of the RyRs constrained to a 20-nm-wide cylindrical shell surrounding the t-tubules. Dilation was performed with a structuring element with a square connectivity of one for nine iterations. This method was chosen because it produces a jSR that evenly surrounds the RyR channels in the cluster, defines a well-constrained dyadic cleft space, and has a jSR volume on the order of $3–4 \times 10^{-12}$ µL, which is in good agreement with the values reported by electron microscopy tomography (*Hake et al., 2012*). The non-junctional network SR (nSR) was added as a regular grid of thin structures throughout the cytosol. While the generated morphology is artificial, the created structure has a volume and surface area in agreement with 3D electron microscopy data (*Hake et al., 2012*) and serves to connect the different CRUs.

For each CRU in the fully reconstructed volumetric geometry, a surrounding region measuring 1 µm × 1 µm × 1 µm was extracted, producing a set of smaller geometries representing individual CRUs for computational modeling of $Ca^{2+}$ release. Roughly 1000 such CRU geometries were extracted and sorted based on the number of contained RyRs, their spatial spread (measured as the root mean square of their distance to the cluster center), and the number of individual clusters within the CRU. Four representative geometries were selected containing roughly the same number of RyRs, but with different degrees of compactness. These ranged from a completely solid CRU to a majorly dispersed CRU consisting of numerous, sprawled clusters (*Supplementary file 1*).

## $Ca^{2+}$ imaging and analysis

Using an LSM 7Live confocal microscope (Zeiss), spontaneous $Ca^{2+}$ sparks were recorded from quiescent cardiomyocytes loaded with fluo-4 AM (20 µmol/L, Molecular Probes, OR) and superfused with HEPES-Tyrode solution containing (in mmol/L): 140 NaCl, 0.5 $MgCl_2$, 5.0 HEPES, 5.5 glucose, 0.4 $NaH_2PO_4$, 5.4 KCl, and 1.8 $CaCl_2$ (pH 7.4, 37°C). Cardiomyocytes were scanned along a 1024 pixel line drawn across the cell's longitudinal axis at a temporal resolution of 1.5 ms for a total duration of 6 s. $Ca^{2+}$ sparks were analyzed using a custom program (CaSparks 1.01, D. Ursu, 2003) as previously described (*Louch et al., 2013*), with sparks defined as local increases in fluorescence intensity of at least three times above background ($\Delta F/F_0 \geq 0.3$). Spark characteristics outputted by the program included amplitude ($\Delta F/F_0$), TTP, full duration at half maximum (FDHM), and full width at half maximum (FWHM). Spark frequency was determined by normalizing spark count to the cell length and recording duration. Spark-mediated leak was calculated as the product of spark amplitude, FDHM, and FWHM.

$Ca^{2+}$ transients were similarly recorded by confocal line-scans in fluo-4-loaded cells, but during field stimulation via a pair of platinum electrodes (3 ms supra-threshold current pulses at 1 Hz). As previously described (*Louch et al., 2006*), $Ca^{2+}$ release synchrony was calculated by plotting the profile of time to 50% peak fluorescence ($TTF_{50}$) across the cell and measuring the standard deviation of the values. We have termed this measure the 'dyssynchrony index.' Spontaneous $Ca^{2+}$ waves were measured during pauses in the electrical excitation, and wave frequency was normalized to the recording duration. SR $Ca^{2+}$ content was estimated by measuring the increase in cytosolic $Ca^{2+}$ fluorescence after superfusing cardiomyocytes with 10 mM caffeine (Sigma-Aldrich). SERCA activity was estimated as the difference between the rate constants ($1/\tau$) of the decline of steady-state $Ca^{2+}$ transient (1 Hz) and the caffeine-induced $Ca^{2+}$ transient ($1/\tau$ 1 Hz − $1/\tau$ caffeine) acquired in the same cell (*Mørk et al., 2009*).

## Western blotting

Frozen tissue from rat left ventricles was homogenized in cold buffer (210 mM sucrose, 2 mM EGTA, 40 mM NaCl, 30 mM HEPES, 5 mM EDTA) with the addition of a cOmplete EDTA-free protease inhibitor cocktail tablet (Roche Diagnostics, Oslo, Norway) and a PhosSTOP tablet (Roche). SDS was then added to the homogenates to a final concentration of 1%, and protein concentrations were quantified using a micro BCA protein assay kit (Thermo Fisher Scientific Inc, Rockford, IL). BSA was used as standard protein.

Protein homogenates (5 or 15 µg/lane) were size-fractionated on 4–15% or 15% Criterion TGX gels (Bio-Rad Laboratories, Oslo, Norway) and transferred to 0.45 µM PVDF membranes (GE Healthcare). The membranes were blocked in 5% non-fat milk or 5% Casein (Roche Diagnostics) in Tris-buffered saline with 0.1% Tween (TBS-T) for 1 hr at room temperature, and then incubated with primary antibody overnight at 4°C. The following primary antibodies were employed for immunoblotting: RyR2 (1:1000) (MA3-916, Thermo Fisher Scientific), pSer2808 RyR2 (1:2500) (A010-30, Badrilla), pSer2814 RyR2 (1:2500) (A010-31, Badrilla), and GAPDH (1:500; sc-20357, Santa Cruz Biotechnology). Secondary

antibodies were anti-rabbit (NA934V, GE Healthcare) or anti-mouse (NA931V, GE Healthcare). These were incubated for 1 hr at room temperature, and blots were developed using Enhanced Chemiluminescence (ECL prime, GE Healthcare). Chemiluminescence signals were detected with a LAS 4000 (GE Healthcare), and protein levels were quantified using ImageQuant software (GE Healthcare). Results were normalized to GAPDH and then to sham values.

## Flow cytometry using fluorescence-activated cell sorting

In order to examine the relative phosphorylation levels of isoproterenol-treated cardiomyocytes, we employed fluorescence-activated cell sorting. This technique has previously been shown to enable protein quantification in close correlation with results from Western blotting (*Krutzik and Nolan, 2003*). Fixed cells labeled with primary antibodies against pSer-2808 RyR2 (A010-30, Badrilla) or pSer-2814 RyR2 (A010-31, Badrilla) and secondary Alexa Fluor 647 antibody were placed in a suspension of PBS with 1% BSA. Flow cytometry analysis was performed on a Sony SH800 Cell Sorter, and FlowJo software (Becton, Dickinson and Company, Ashland, OR) was used for analysis. Control treatments were included, which contained unlabeled cells or only secondary antibody. Voltage settings for the side, back, and forward scatter were kept constant for each experiments described. The cell suspension was analyzed at a flow rate of 27 µL/min, until >5000 events were captured. A two-step sorting process was employed. Intact cardiomyocytes were separated from other cell types and debris using a dual-parameter dot plot for side and forward scatter (see *Figure 2—figure supplement 1a* for representative image). Single cardiomyocytes were separated from doublets using a plot of forward scatter height and pulse width (*Figure 2—figure supplement 1b*).

Following sorting, the secondary Alexa Fluor 647 antibody (A-21237, Thermo Fisher Scientific) in the single cardiomyocyte fraction was excited with the 638 nm laser and emission detected by a photomultiplier tube with a 665/30 band-pass filter. Data are presented as detected fluorescence intensity as a product of cell size. Confirmation of cell viability was performed using LIVE/DEAD Fixable Violet Dead Cell Stain (Thermo Fisher), excited with the 405 laser and emission detected via a 450/50 band-pass filter.

## Mathematical modeling of $Ca^{2+}$ sparks

Effects of changing CRU configuration on $Ca^{2+}$ spark characteristics were interrogated using a mathematical model of $Ca^{2+}$ release in the dyad. This previously described reaction-diffusion model includes independent, stochastic RyR channels coupled with deterministic $Ca^{2+}$ diffusion and buffering (*Kolstad et al., 2018*). The model was chosen for the current work since it can incorporate changing CRU configurations based on super-resolution imaging. The model was employed as previously described, but with minor adjustments to better align $Ca^{2+}$ release and pump-leak balance behavior with experimental data from rat and mouse (*Mohamed et al., 2018*; *Santiago et al., 2013*; *Shannon et al., 2002*). Specifically, we slightly augmented SR $Ca^{2+}$ reuptake by increasing the amount and density of SERCA, and modestly decreased SR volume and buffering to reduce releasable $Ca^{2+}$. When combined with a slightly more sensitive RyR model (*Laver et al., 2017*), these modifications yielded larger fractional SR $Ca^{2+}$ release in better agreement with experimental data. Please refer to *Supplementary file 2* and *Supplementary file 3* for a full list of buffering and RyR parameters.

To incorporate the effects of isoproterenol treatment on $Ca^{2+}$ signaling in the CRU, we altered two model parameters: the SR $Ca^{2+}$ load was increased from 900 µM at baseline to 1300 µM, and the RyR opening sensitivity to cytosolic $Ca^{2+}$ was increased by lowering the $K_d$ open from 45 µM to 25 µM. These two model parameters and their adjustments were constrained by comparing the amplitude and rise time of simulated sparks to experimental measurements made in control and isoproterenol treatment conditions (*Figure 5—figure supplement 2*).

## Analysis of spark simulations

A total of 400 simulations were performed for both baseline and isoproterenol parameter sets, with each CRU geometry. The initial conditions were otherwise identical for each simulation. A single, randomly selected RyR in the CRU was opened, and the simulation was then allowed to progress stochastically. The probability of each RyR opening and closing was thus based on its own locally sensed $Ca^{2+}$ concentration. The simulations continued until all RyRs in the CRU had been simultaneously

closed for at least 1 ms, at which point the simulation was terminated. Thus, the full tail of the spark was not modeled out of consideration for computational efficiency.

To compare the model with experimentally measured sparks, we used the average concentration of $Ca^{2+}$-bound Fluo in the full cytosol of the 1 μm × 1 μm × 1 μm computational domain. This measure was employed instead of more computationally expensive line-scan simulations since our previous work indicated that this simplified approach yields equal predictions of spark amplitude and time course (*Kolstad et al., 2018*).

For each simulation, we measured the peak amplitude of the Fluo signal and the rise time (TTP). Based on the measured amplitude, we then determined which simulations to consider sparks using the same threshold as in the experimentally measured $Ca^{2+}$ sparks ($\Delta F/F_0 \geq 0.3$). For simulations where the peak amplitude was below this threshold, we defined sub-spark events as 'quarks' (*Brochet et al., 2011*) ($0.1 < \Delta F/F_0 < 0.3$) or 'failed sparks' if $Ca^{2+}$ release was negligible ($\Delta F/F_0 \leq 0.1$).

Based on the ratio of spark events to sub-spark events, we estimated the probability of a spontaneous opening leading to an observable spark, dubbed the 'spark fidelity.' As each individual simulation was an independent stochastic trial in a Bernoulli process, each simulation produced a spark with probability equal to the fidelity. To estimate the fidelity, we used the maximum-likelihood predictor of the probability, which is simply the observed ratio of successful sparks to total simulations. Extra care should be taken while calculating a confidence interval for spark fidelity as we might expect the fidelity to be very close to or equal to 0 for severely fragmented clusters. We therefore opted to follow the recommendation of *Brown et al., 2017* and used the Agresti–Coull confidence interval (*Agresti and Coull, 1998*).

We also quantified the amount of released $Ca^{2+}$ during each simulation. For simulations that were deemed to elicit observable sparks, the released $Ca^{2+}$ was considered spark-mediated leak, while sub-spark events defined 'silent' leak. Total RyR-mediated leak was calculated as the sum of the two types of events. As the mathematical model was mainly constrained using amplitude and TTP values of sparks, the leak measurements were not analyzed in absolute terms, but rather as the relative amount of spark-mediated and silent leak in the different geometry and parameter combinations.

## Statistical analyses

Cardiomyocytes from rats were randomly selected for analysis of RyR localization and $Ca^{2+}$ imaging. Power analysis was performed to determine sample sizes based on known variability of measured parameters. No data were excluded from the analyses, and consistent observations were made during analyses performed on different cardiomyocytes from different hearts. Experiments involving multiple treatment groups were all carried out on the same day to ensure consistency. All results presented are expressed as mean ± standard error of the mean unless otherwise stated. For RyR localization and $Ca^{2+}$ imaging experiments, the presented bar charts include data points that are averaged measurements from each individual cardiomyocyte. For Western blotting results, data points are presented from each heart examined. Statistical significance between sample means was calculated by nested ANOVA (SPSS, IBM), with Fisher's least significant difference post-hoc comparison where appropriate. p-Values<0.05 were considered to be significant.

## Code availability

Custom code used in this study is available at the public repository https://gitlab.com/louch-group/ryr-tt-correlative-analsyis (*van den Brink et al., 2022*).

## Acknowledgements

We thank the Section for Comparative Medicine at Oslo University Hospital Ullevål for expert animal care. This study was financially supported by the European Union's Horizon 2020 research and innovation program (consolidator grant, WEL) under grant agreement no. 647714, the Norwegian Research Council (XS, WEL), and the Norwegian Association for Public Health (WEL). This work used the Oakforest-PACS supercomputer system provided by the University of Tokyo through Joint Usage/Research Center for Interdisciplinary Large-scale Information Infrastructures and High Performance Computing Infrastructure in Japan (project IDs: JHPCN-jh180024, JHPCN-jh190040, and JHPCN-jh200036).

# Additional information

## Funding

| Funder | Grant reference number | Author |
|---|---|---|
| Norwegian Research Council | 287395 | Xin Shen<br>Yufeng Hou<br>Martin Laasmaa |
| European Research Council | 647714 | Xin Shen<br>Terje R Kolstad<br>William E Louch |

The funders had no role in study design, data collection and interpretation, or the decision to submit the work for publication.

## Author contributions

Xin Shen, Conceptualization, Resources, Data curation, Software, Formal analysis, Supervision, Validation, Investigation, Visualization, Methodology, Writing – original draft, Project administration, Writing – review and editing; Jonas van den Brink, Conceptualization, Resources, Data curation, Software, Formal analysis, Validation, Methodology, Writing – original draft, Writing – review and editing; Anna Bergan-Dahl, Data curation, Formal analysis, Validation, Visualization, Writing – original draft, Writing – review and editing; Terje R Kolstad, Resources, Formal analysis, Validation, Investigation, Visualization, Methodology, Writing – review and editing; Einar S Norden, Resources, Validation, Investigation, Methodology, Writing – review and editing; Yufeng Hou, Resources, Data curation, Software, Formal analysis, Validation, Visualization, Methodology, Writing – review and editing; Martin Laasmaa, Resources, Data curation, Software, Formal analysis, Validation, Visualization, Writing – review and editing; Yuriana Aguilar-Sanchez, Resources, Data curation, Validation, Methodology; Ann P Quick, Resources, Data curation, Validation, Methodology, Writing – review and editing; Emil KS Espe, Ivar Sjaastad, Resources, Data curation, Validation, Investigation, Visualization, Methodology, Writing – review and editing; Xander HT Wehrens, Resources, Supervision, Validation, Writing – review and editing; Andrew G Edwards, Conceptualization, Resources, Software, Supervision, Validation, Investigation, Visualization, Writing – review and editing; Christian Soeller, Conceptualization, Resources, Supervision, Validation, Investigation, Visualization, Methodology, Writing – review and editing; William E Louch, Conceptualization, Resources, Supervision, Funding acquisition, Validation, Writing – original draft, Project administration, Writing – review and editing

## Author ORCIDs

Xin Shen http://orcid.org/0000-0003-4429-8358
Terje R Kolstad http://orcid.org/0000-0002-0589-5689
Martin Laasmaa http://orcid.org/0000-0002-6663-6947
Christian Soeller http://orcid.org/0000-0002-9302-2203
William E Louch http://orcid.org/0000-0002-0511-6112

## Ethics

All animal experiments were performed in accordance with the Norwegian Animal Welfare Act and NIH Guidelines, and were approved by the Ethics Committee of the University of Oslo and the Norwegian animal welfare committee (FOTS ID 20208). The majority of the experiments were performed on adult male Wistar rats (250–350 g) purchased from Janvier Labs (Le Genest-Saint-Isle, France). Rats were group -housed at 22°C on a 12 hr:12 hr light–dark cycle, with free access to food and water. Cardiomyocytes isolated from transgenic RyR2-S2814D and RyR2-S2814A mice were provided by the laboratory of Xander Wehrens (Baylor College of Medicine, Texas, United StatesTX), where experiments were performed in accordance with the Guide for the Care and Use of Laboratory Animals (National Academies Press, 2011) and approved by the Baylor College of Medicine Institutional Animal Care and Use Committee. A total of 64 rats and 6 mice were used in this study.

## Decision letter and Author response
Decision letter https://doi.org/10.7554/eLife.77725.sa1
Author response https://doi.org/10.7554/eLife.77725.sa2

# Additional files

## Supplementary files

- Supplementary file 1. Definition of modeled ryanodine receptor (RyR) cluster geometries. From a large number of automatically generated 3D Ca$^{2+}$ release unit (CRU) geometries, four were selected to be used in mathematical modeling of Ca$^{2+}$ release. These were chosen to represent the range of fragmentation observed experimentally, while ensuring that each CRU contained roughly the same number of RyRs. Characteristics for each CRU are listed. Spatial spread among the RyRs within each geometric configuration was summarized with calculation of the root mean square distance of RyRs to the CRU center.

- Supplementary file 2. Model parameters for Ca$^{2+}$ buffering. Ca$^{2+}$ diffusion is subject to mobile and immobile buffers in the cytosolic and sarcoplasmic reticulum (SR) compartments and was modeled for five buffer species (left) in kinetic detail. Only calsequestrin was modeled inside the SR domain, while the remaining buffers were cytosolic. Troponin and calsequestrin were modeled as immobile ($\sigma = 0$). The buffering capacity of each buffer species is defined by $B_{tot}$, and the binding on- and off-rates are given by $k_{on}$ and $k_{off}$, respectively.

- Supplementary file 3. Ryanodine receptor (RyR) model parameters. Parameters of the two-state RyR model. The on-rate (+) and off-rate (-) for Ca$^{2+}$ activation are given in the two rows of the table. For each rate, $k_{min}$ and $k_{max}$ represent the upper and lower bounds at very low and high cytosolic calcium ($[Ca^{2+}]_i$). These rate bounds apply when the corresponding value for $([Ca^{2+}]_i/K_d)^n$ is outside of the bounds they set. Otherwise, both rates are calculated via the corresponding $([Ca^{2+}]_i/K_d)^n$. The on-rate was modified to simulate sensitization to cytosolic Ca$^{2+}$ accompanying β-adrenergic receptor (β-AR) stimulation by shifting the half-maximal concentration from 45 μM to 25 μM, as denoted by *.

- Supplementary file 4. Cardiac parameters from post-myocardial infarction (MI) rats with heart failure (HF) and sham-operated controls. HF development was examined by CINE MRI assessment based on established criteria; *Sjaastad et al., 2000*. Infarct size is percentage of left ventricular free wall (sham: $n_{heart}$ = 14; HF: $n_{heart}$ = 12; *p<0.05).

- Transparent reporting form

## Data availability

Custom codes used in this study were written in Python, and are available in the public repository https://gitlab.com/louch-group/ryr-tt-correlative-analsyis, (copy archived at swh:1:rev:2524ae78b650c2429991e53d26a885afedf321bd).

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
