## [Editor Report]

This article is of fundamental interest to our understanding of heart failure and the cardiac contraction process. It applies super-resolution imaging and functional calcium imaging to healthy and failing cardiac cells and combines these with quantitative modeling of the resulting data. The outcome bearing on receptor distribution is a good example of exploiting quantitative super-resolved data in combination with other techniques to gain real insight into a biological problem.

---

## [Decision Letter]

**Decision letter after peer review:**

Thank you for submitting your article "Prolonged β-Adrenergic Stimulation Disperses Ryanodine Receptor Clusters in Cardiomyocytes: Implications for Heart Failure" for consideration by *eLife*. Your article has been reviewed by 2 peer reviewers, and the evaluation has been overseen by a Reviewing Editor and Richard Aldrich as the Senior Editor. The reviewers have opted to remain anonymous.

The reviewers have discussed their reviews with one another, and the Reviewing Editor has drafted this to help you prepare a revised submission. Although a significant number of suggestions are made, these are meant to be constructive towards the enhancement of your paper.

Essential revisions:

The reviewers make the following broad points:

Reviewer 1 would like you to consider the following suggestions:

First, in the absence of a control with RyR constitutively active at the S2808 (PKA) phosphorylation site, it would be interesting to see if direct activation of PKA using 6MB-cAMP is sufficient to induce disruption of the RyR-S2814A. Conversely, the ability of PKA inhibition to reverse the ISO-induced RyR disruption of RyR-S2814D could also be tested.

Along similar lines, after prolonged stimulation of β-AR, or in failing cardiomyocytes inhibition of either CaMKII or PKA partially reverses RyR disruption and CICR disruption. Have you tested whether the disruption is completely reversed by the simultaneous inhibition of both?

The link between β-AR stimulation and phosphorylation-dependent RyR dispersion in heart failure is assumed rather than tested (line 393). I note that care has been taken not to assert this too strongly and that phosphorylation-dependent RyR dispersion is the key claim, but I am curious to know if considered any ways to test it (perhaps by running experiments in post-infarction cells where β-AR has been inhibited)? I leave this as an open question as it isn't clear to me how best to do this and it is perhaps not trivial, possible, or necessary!

Reviewer 2 makes the comments that:

– A critical control is missing to demonstrate the effects of 2 hrs ISO treatment on RyR2 arrangements. This will strengthen the conclusions from the CaMKII and PKA inhibitor RYR2 dispersion reversal experiments which utilize a protocol that includes a 1 hr ISO stimulation followed by a 1Hr ISO + inhibitor treatment.

– The mathematical modeling results would also benefit from more explanation of how they complement, align, and expand on the experimental studies.

The reviewers also make the following major points:*Reviewer #1 (Recommendations for the authors):*

1. A brief explanation of how CRU fragmentation would be reflected by changes to the reported dSTORM parameters (RyR cluster size, CRU size, and cluster density) at the start of the Results section would help to orient the reader.

2. Reconstructions of representative CRUs are presented in Figure 1, 2, and 3, with selected CRUs inset as compelling visual illustrations of CRU fragmentation or reformation. It would be beneficial to allow the reader to decide for themselves if they are truly representative by stating their dSTORM parameter values in the figure or legend. This is especially true where single representative CRUs are presented as in Figure 2a-c and Figure 3a and b.

3. On line 192 the authors state 'We found that the RyR arrangement in the phosphomimetic S2814D 193 mutant cells at baseline resembled the fragmented RyR organization observed in wild type (WT) 194 animals after isoproterenol treatment (Figure 2g, h). Indeed, cluster size (Figure 2j), CRU size (Figure 2k), and 195 cluster density (Figure 2l) were all similar in WT + ISO and S2814D cardiomyocytes, and significantly 196 different from untreated WT cells. Furthermore, application of isoproterenol to S2814D cardiomyocytes 197 did not lead to significant additional RyR dispersion, and only tendencies toward smaller cluster and 198 CRU sizes (Figure 2h, j-l).'

The representative CRU reconstructions in Figure 2g and h seem to contradict the quantified dSTORM parameters in Figure 2h, j-l and the description of the results in the text (starting at Line 192). This indicates a possible error in the production of the figure or that the choice of representative CRUs could be improved. In the images, the S214D CRUs appear more similar to wt (aside from the CRU selected for the insert) rather than wt + ISO as expected from the plotted data. There also appears to be significantly greater disruption visible in the S2814D + ISO than S2814D as a result, again counter to the observation made in the text about the plotted data.

4. There is a minor discrepancy between the naming of CaMKII mutants in Figure 2. panels (2814D and 2814A) and the text (S2814D and S2814A).

5. A more precise description of the source of the S2814D and S2814A cardiomyocytes is needed for the reader to fully understand the strength of these control experiments. The authors currently state in the text and Materials and methods section that they are isolated from transgenic RyR2-S214D and RyR2-S214A mice provided by the Wehrens lab. The paper reference provided is only partially helpful as it describes the generation of homozygous knock-in mice RyR2-S214A but makes no mention of RyR2-S214D. I would assume that both are isolated from homozygous knock-in mice but please confirm if this is indeed the case and state it explicitly in the manuscript.

6. Figure 2, 3, 4, and 6 legends refer to various bar charts of quantified dSTORM parameters and ca^2+^ spark and wave properties as 'Mean data' and it is not clear what this means. It might be less confusing simply to use the term 'Experimental data' instead.

7. Connected to the previous point, the bar charts in Figure 1, 2, 3, 4, and 6 are nicely presented so that the mean, SEM, and variation in the datapoints of a given variable are clear. However, it is not always obvious whether each datapoint represents individual measurements or a cell average and this information should be stated clearly in either the legends or methods section.

8. A brief description of how spark-mediated ca^2+^ leak is calculated from spark mass and frequency is missing from the Materials and methods section.

9. Figure 5c-f: These panels seem to have shared keys for the data represented by unshaded and shaded bars that only appear in the final panel (Figure 5f). It would be clearer if the difference between unshaded and shaded was described in the figure legend with reference to the applicable panels. The same applies to Figure 6f-g.

10. The Ethical approval section of Materials and methods needs to be checked for references to other journals.

11. Supplementary Tables 1-3 would benefit from brief explanations of their contents.*Reviewer #2 (Recommendations for the authors):*

1) The authors report that a 1 hr ISO stimulation followed by a 1 hr ISO stimulation in the presence of AIP led to a reversal of the ISO-induced RyR2 dispersal and concluded that CaMKII thus mediated RyR2 cluster dispersion downstream of B-AR stimulation. However, in this protocol, the cells are fixed after a 2 hr total exposure to ISO and it is unknown what this duration of ISO treatment does to RyR2 clustering and distribution. To increase confidence in these findings, an important control must be added. RyR2 clusters should be examined after 2 hrs ISO treatment (without any inhibitors on board). Likewise, a similar protocol is employed to test the effects of PKA on cluster dispersal, with the authors reporting that H89 reversed the dispersal. Again, this finding would be strengthened by knowing what 2 hrs of ISO does to RyR2 clustering and distribution. The same 2 hr total ISO stimulation protocol is used when the authors assess whether H89 or AIP can reverse the slowing in spark kinetics. To strengthen confidence in this finding, spark kinetics should also be examined after 2hrs of ISO treatment. The results may be quite different from those seen after a 1 hr treatment especially given the finding that RyR phosphorylation at ser-2814 and ser-2808 both exhibited bell-shaped responses during the 1 h ISO stimulation. If one extends that treatment to 2 hrs, phosphorylation of S2814 and S2808 may continue to fall and this complicates the findings with the inhibitors.

2) At each ISO-treated time point, the study would benefit from the inclusion of an accompanying no-ISO control. Does the RyR dispersal occur because the cells are deteriorating with more time post-isolation, or is it specifically and exclusively because of the acute ISO-induced stress? These controls would help address that question and strengthen the conclusions of the paper.

3) There is no indication of statistical significance on any of the summary data plots or mention of P values. This should be addressed.

4) The modeling results need to be put into context and compared with what is going on in the experimental data. This section is somewhat confusing and is a bit of a jarring shift away from the experimental section. It should be better integrated. The authors state "the modeling results indicate that RyR ca^2+^ leak shifts from spark-mediated to silent leak when CRUs are dispersed, reducing the fidelity of spark generation" but in figure 4h, spark-mediated (non-silent) leak is strikingly increased after 60 mins of ISO treatment which leads to the most profound dispersal. Please clarify how these results align. Perhaps there is more of both spark-mediated leak and non-spark leak? What experimental data supports non-spark mediated leak? The idea that there could be localized regions of SR ca^2+^ depletion that act as "fire-breaks" to prevent the generation of ca^2+^ waves even though there are more ca^2+^ sparks is logical and provides a satisfying theory that unifies some of the seemingly paradoxical findings but this is only raised in the second to last paragraph of the discussion. Raising this idea earlier may help the reader connect the dots more easily.

---

## [Author Response]

Essential Revisions (for the authors):The reviewers make the following broad points:Reviewer 1 would like you to consider the following suggestions:First, in the absence of a control with RyR constitutively active at the S2808 (PKA) phosphorylation site, it would be interesting to see if direct activation of PKA using 6MB-cAMP is sufficient to induce disruption of the RyR-S2814A. Conversely, the ability of PKA inhibition to reverse the ISO-induced RyR disruption of RyR-S2814D could also be tested.

Thank you for these suggestions for these additional control experiments, which we have now performed. As expected, 6MB-cAMP remained effective in dispersing RyR clusters in RyR-S2814A cells, supporting that 6MB exerts its actions independent from the CaMKII phosphorylation site on RyR. For the second experiment, we believe you meant to suggest applying PKA inhibition (H89) together with ISO treatment in the S2814A cells, not S2814D cells. This experiment would then show that H89-induced inhibition of RyR dispersion is mediated via PKA inhibition without contribution of the 2814 site. Performing these experiments confirmed this hypothesis. We have added these new data to the bottom of Figure 3, and included the following addition to the main text:

Page 4, line 228:

“Importantly, 6MB-induced RyR dispersion in S2814A cardiomyocytes (Figure 3f, h-j) which was comparable to that observed in WT. Similarly, dispersion observed in S2814A cardiomyocytes treated with isoproterenol was reversed by the addition of H89 (Figure 3g-j). These observations support that 6MB and H89 treatments exerted their effects on RyR configuration via modulation of PKA activity, without contribution from the CaMKII phosphorylation site on the RyR. Taken together, these data indicate that prolonged PKA-dependent phosphorylation of RyRs is sufficient to drive RyR cluster fragmentation, but that both PKA and CaMKII concertedly drive this process during β-AR stimulation.”

Along similar lines, after prolonged stimulation of β-AR, or in failing cardiomyocytes inhibition of either CaMKII or PKA partially reverses RyR disruption and CICR disruption. Have you tested whether the disruption is completely reversed by the simultaneous inhibition of both?

The reviewer is correct that inhibition of CaMKII or PKA individually did not fully reverse the effects of ISO treatment on RyR configuration and ca^2+^ sparks. We have now tested the combined actions on AIP and H89 in ISO-treated cells. As expected, we observed a significant further reversal of the RyR dispersion induced from ISO treatment (Figure 3 —figure supplement 5a-c). Similarly, we observed a tendency for further reversal of spark characteristics when AIP and H89 were combined (Figure 3 —figure supplement 5d-f).

We have added text accordingly to two parts of the manuscript:

Page 3, Line 216:

“As in experiments investigating effects of CaMKII inhibition (Figure 2), it should be noted that PKA inhibition did not fully restore the RyR configuration at the CRU level. Indeed, simultaneous treatment of cardiomyocytes with AIP and H89 resulted in further reversal of the effects of isoproterenol on CRU size, beyond those observed with either agent alone (Figure 3 —figure supplement 1a-c). These observations are consistent with summative roles of PKA and CaMKII activation in promoting RyR dispersion”

Page 4, Line 252:

“A tendency toward additive effects was noted when AIP and H89 treatments were combined (Figure 3 —figure supplement 1d-f). In control experiments, a 2 h treatment period with isoproterenol alone (matching the total treatment time in AIP and H89 experiments) did not alter ca^2+^ spark properties beyond values observed at the 1 h time point (Figure 2 —figure supplement 2c, d). Taken together, these data indicate that, during prolonged β-AR stimulation, RyR dispersion induced by PKA and CaMKII activity is linked to slowing of ca^2+^ spark kinetics and increased ca^2+^ leak”

The link between β-AR stimulation and phosphorylation-dependent RyR dispersion in heart failure is assumed rather than tested (line 393). I note that care has been taken not to assert this too strongly and that phosphorylation-dependent RyR dispersion is the key claim, but I am curious to know if considered any ways to test it (perhaps by running experiments in post-infarction cells where β-AR has been inhibited)? I leave this as an open question as it isn't clear to me how best to do this and it is perhaps not trivial, possible, or necessary!

The Reviewer is correct in stating that the link between β-AR stimulation and phosphorylation-dependent RyR dispersion in heart failure is only assumed in our study. Although considerable previous work has supported the presence of a hyper-sympathetic state in heart failure, including in post-infarction rats (Ganguly et al., AJP 1997), we have not tested effects of reducing sympathetic tone on RyR configuration. We have now clarified this point, and edited the following sentence in the Results section (Page 7, line 372):

“The striking similarity of these findings to experiments employing prolonged isoproterenol exposure (Figures 1-4) supports that fragmentation of RyR clusters during HF results from increased activation of both CaMKII and PKA, likely as a consequence of the chronic β-AR stimulation that is well established in this condition (Ganguly et al., AJP, 1997).”

We have also raised this point in the Discussion, where we mention the importance of examining effects of β-blockers in future heart failure studies (page 9, line 480):

“Future work should be aimed at directly linking β-adrenergic tone, RyR dispersion, and ca^2+^ signaling impairments in experimental HF, and establish the reversibility of this relationship by the application of β-blockers.”

Reviewer 2 makes the comments that:– A critical control is missing to demonstrate the effects of 2 hrs ISO treatment on RyR2 arrangements. This will strengthen the conclusions from the CaMKII and PKA inhibitor RYR2 dispersion reversal experiments which utilize a protocol that includes a 1 hr ISO stimulation followed by a 1Hr ISO + inhibitor treatment.

Your point is well taken that the total treatment time for ISO + AIP (Figure 2) or ISO + H89 (Figure 3) is 2 hours. Thus, we have now included a 2 h ISO control, and observed no further RyR cluster dispersion beyond levels present in the 1 h ISO group. These data have been presented in Figure 2 —figure supplement 2a, b rather than the main figures which were becoming quite cumbersome. The follow edit has been made to the text on page 3, line 179:

“In control experiments, the longer 2 h treatment period with isoproterenol alone did not alter RyR configuration beyond values observed at the 1 h time point (Figure 2 —figure supplement 2a, b).”

Similarly, we have included 2 hr ISO control experiments for ca^2+^ spark characteristics, and again observed no further effects beyond changes observed at 1 h (Figure 2 —figure supplement 2c, d). Text has been added on page 4, line 253 stating:

“In control experiments, a 2 h treatment period with isoproterenol alone (matching the total treatment time in AIP and H89 experiments) did not alter ca^2+^ spark properties beyond values observed at the 1 h time point (Figure 2 —figure supplement 2c, d).”

– The mathematical modeling results would also benefit from more explanation of how they complement, align, and expand on the experimental studies.

Thank you for this comment. We have now edited the description of the mathematical modeling data, and emphasized how these results complement and expand upon experimental findings. Importantly, we have noted how calculations of spark-mediated and silent leak observed in modeling contribute to reduction in SR ca^2+^ content, and thereby inhibition of ca^2+^ waves following RyR dispersal. Finally, we have included a new paragraph at the end of the modeling Results section summing up our key findings (page 6, line 340):

“Taken together, the modeling results critically link changes in CRU configuration and spark morphology; phenomena that were observed separately in experiments. However, the modeling data also support the contention from experimental data that RyR cluster fragmentation can reduce arrhythmogenic spontaneous ca^2+^ release (ca^2+^ waves). Here, the antiarrhythmic actions of RyR dispersion appear to be linked to a reduction in SR content resulting from an increase in both spark-mediated and silent leak. We postulate that local regions of the cell with dispersed CRUs and lowered SR content may act as a physical barrier or “fire break” to impede ca^2+^ wave propagation.”

The reviewers also make the following major points:Reviewer #1 (Recommendations for the authors):1. A brief explanation of how CRU fragmentation would be reflected by changes to the reported dSTORM parameters (RyR cluster size, CRU size, and cluster density) at the start of the Results section would help to orient the reader.

Your point is well taken, as these changes may be confusing to readers not familiar with previous work describing CRU dispersion. We have now included a brief explanation of how this phenomenon is expected to influence the number of RyRs per cluster and CRU, and the overall cluster and RyR densities, as well as an additional schematic (Figure 1 —figure supplement 1) to illustrate hallmarks of RyR cluster dispersion. The following text has been added to the beginning of the Results section on page 2, line 129:

“Using 3D dSTORM imaging, we investigated the effects of long-term phosphorylation on RyR organization in cardiomyocytes. We specifically hypothesized that phosphorylation would fragment or “disperse” RyR clusters in a time-dependent manner. Numerically, such breaking apart of RyR groupings is expected to be reflected by a reduction in the average number of RyRs contained in each cluster, as illustrated in Figure 1 —figure supplement 1. This change would be accompanied by a concomitant increase in the total number of clusters as there now more numerous, smaller clusters in a given 3D space. If marked RyR cluster dispersion occurs, the total number of RyRs contained in a CRU may also be reduced, as clusters are no longer located in close enough proximity to be grouped in functional release units (edge-to-edge distances ≤ 100 nm; Figure 1 —figure supplement 1). Notably, if RyR dispersion occurs without the addition or loss of channels, then the total number of RyRs detected by dSTORM imaging should remain unchanged.”

2. Reconstructions of representative CRUs are presented in Figure 1, 2, and 3, with selected CRUs inset as compelling visual illustrations of CRU fragmentation or reformation. It would be beneficial to allow the reader to decide for themselves if they are truly representative by stating their dSTORM parameter values in the figure or legend. This is especially true where single representative CRUs are presented as in Figure 2a-c and Figure 3a and b.

We have now added parameter values below each of the representative CRUs in Figures 1, 2 and 3. We hope that this gives the reader a feeling for how the example images compare with mean observed changes in RyR configuration.

3. On line 192 the authors state 'We found that the RyR arrangement in the phosphomimetic S2814D 193 mutant cells at baseline resembled the fragmented RyR organization observed in wild type (WT) 194 animals after isoproterenol treatment (Figure 2g, h). Indeed, cluster size (Figure 2j), CRU size (Figure 2k), and 195 cluster density (Figure 2l) were all similar in WT + ISO and S2814D cardiomyocytes, and significantly 196 different from untreated WT cells. Furthermore, application of isoproterenol to S2814D cardiomyocytes 197 did not lead to significant additional RyR dispersion, and only tendencies toward smaller cluster and 198 CRU sizes (Figure 2h, j-l).'The representative CRU reconstructions in Figure 2g and h seem to contradict the quantified dSTORM parameters in Figure 2h, j-l and the description of the results in the text (starting at Line 192). This indicates a possible error in the production of the figure or that the choice of representative CRUs could be improved. In the images, the S214D CRUs appear more similar to wt (aside from the CRU selected for the insert) rather than wt + ISO as expected from the plotted data. There also appears to be significantly greater disruption visible in the S2814D + ISO than S2814D as a result, again counter to the observation made in the text about the plotted data.

We agree with the Reviewer that our previous choice of representative image for Figure 2h did not accurately reflect the mean characteristics of RyR configurations in the 2814D cells. We have now selected a more suitable example. The appropriateness of each representative example is now documented by the presentation of the RyR cluster and CRU measurements below each image (see previous comment).

4. There is a minor discrepancy between the naming of CaMKII mutants in Figure 2. panels (2814D and 2814A) and the text (S2814D and S2814A).

We have corrected the figure legend to read S2814D and S2814A.

5. A more precise description of the source of the S2814D and S2814A cardiomyocytes is needed for the reader to fully understand the strength of these control experiments. The authors currently state in the text and Materials and methods section that they are isolated from transgenic RyR2-S214D and RyR2-S214A mice provided by the Wehrens lab. The paper reference provided is only partially helpful as it describes the generation of homozygous knock-in mice RyR2-S214A but makes no mention of RyR2-S214D. I would assume that both are isolated from homozygous knock-in mice but please confirm if this is indeed the case and state it explicitly in the manuscript.

We have now added the reference for the generation of S2814D mice, and clarified that we employed homozygous S2814A and S2814D animals and their wildtype littermates in our study. The following text has been added to the Methods section on page 10, line 521:

“We examined effects of the CaMKII phosphorylation site S2814 on RyR configuration by employing knock-in transgenic mouse models, where the site was rendered constitutively active (RyR2-S2814D^45^) or genetically ablated (RyR2-S2814A^46^). Cells isolated from homozygous mutant mice were used for this study, with comparison made to their respective wild-type littermates which served as controls.”

6. Figure 2, 3, 4, and 6 legends refer to various bar charts of quantified dSTORM parameters and ca^2+^ spark and wave properties as 'Mean data' and it is not clear what this means. It might be less confusing simply to use the term 'Experimental data' instead.

We agree that this was confusing, and have amended the text to clarify this point throughout the manuscript.

7. Connected to the previous point, the bar charts in Figure 1, 2, 3, 4, and 6 are nicely presented so that the mean, SEM, and variation in the datapoints of a given variable are clear. However, it is not always obvious whether each datapoint represents individual measurements or a cell average and this information should be stated clearly in either the legends or methods section.

We have now clarified this point on page 15, line 792, of the Statistics Analyses section in the Methods:

“All results presented are expressed as mean ± standard error of the mean (SEM) unless otherwise stated. For RyR localisation and ca^2+^ imaging experiments, the presented bar charts include data points which are averaged measurements from each individual cardiomyocyte. For Western blotting results, data points are presented from each heart examined.”

We have also included a sentence in the figure legends which reads:

“The bar charts present mean measurements ± SEM, with superimposed data points representing averaged values from each cardiomyocyte.”

8. A brief description of how spark-mediated ca^2+^ leak is calculated from spark mass and frequency is missing from the Materials and methods section.

Spark-mediated leak was calculated as the product of spark amplitude, FDHM and FWHM. This has now been mentioned on page 13, line 672 of the Methods section.

9. Figure 5c-f: These panels seem to have shared keys for the data represented by unshaded and shaded bars that only appear in the final panel (Figure 5f). It would be clearer if the difference between unshaded and shaded was described in the figure legend with reference to the applicable panels. The same applies to Figure 6f-g.

9. Figure 5c-f: These panels seem to have shared keys for the data represented by unshaded and shaded bars that only appear in the final panel (Figure 5f). It would be clearer if the difference between unshaded and shaded was described in the figure legend with reference to the applicable panels. The same applies to Figure 6f-g.

This has now been clarified in the legend.

10. The Ethical approval section of Materials and methods needs to be checked for references to other journals.

Corrected (page 8, line 509).

11. Supplementary Tables 1-3 would benefit from brief explanations of their contents.

Thank you for requesting this clarification, we have now elaborated on the descriptions for these supplementary tables (now named supplementary files 1-3).

Reviewer #2 (Recommendations for the authors):1) The authors report that a 1 hr ISO stimulation followed by a 1 hr ISO stimulation in the presence of AIP led to a reversal of the ISO-induced RyR2 dispersal and concluded that CaMKII thus mediated RyR2 cluster dispersion downstream of B-AR stimulation. However, in this protocol, the cells are fixed after a 2 hr total exposure to ISO and it is unknown what this duration of ISO treatment does to RyR2 clustering and distribution. To increase confidence in these findings, an important control must be added. RyR2 clusters should be examined after 2 hrs ISO treatment (without any inhibitors on board). Likewise, a similar protocol is employed to test the effects of PKA on cluster dispersal, with the authors reporting that H89 reversed the dispersal. Again, this finding would be strengthened by knowing what 2 hrs of ISO does to RyR2 clustering and distribution. The same 2 hr total ISO stimulation protocol is used when the authors assess whether H89 or AIP can reverse the slowing in spark kinetics. To strengthen confidence in this finding, spark kinetics should also be examined after 2hrs of ISO treatment. The results may be quite different from those seen after a 1 hr treatment especially given the finding that RyR phosphorylation at ser-2814 and ser-2808 both exhibited bell-shaped responses during the 1 h ISO stimulation. If one extends that treatment to 2 hrs, phosphorylation of S2814 and S2808 may continue to fall and this complicates the findings with the inhibitors.

As we have noted above, your point is well taken that the total treatment time for ISO + AIP (Figure 2) or ISO + H89 (Figure 3) is 2 hours. Thus, we have now included a 2 h ISO control, and observed no further RyR cluster dispersion beyond levels present in the 1 h ISO group. These data have been presented in Figure 2 —figure supplement 2a, b rather than the main figures which were becoming quite cumbersome. The following edit has been made to the text on page 3, line 179:

“In control experiments, the longer 2 h treatment period with isoproterenol alone did not alter RyR configuration beyond values observed at the 1 h time point (Figure 2 —figure supplement 2a, b)”

Similarly, we have included 2 hr ISO control experiments for ca^2+^ spark characteristics, and again observed no further effects beyond changes observed at 1 h (Figure 2 —figure supplement 2c, d). Text has been added on page 4, line 253 stating:

“In control experiments, a 2 h treatment period with isoproterenol alone (matching the total treatment time in AIP and H89 experiments) did not alter ca^2+^ spark properties beyond values observed at the 1 h time point (Figure 2 —figure supplement 2c, d).”

2) At each ISO-treated time point, the study would benefit from the inclusion of an accompanying no-ISO control. Does the RyR dispersal occur because the cells are deteriorating with more time post-isolation, or is it specifically and exclusively because of the acute ISO-induced stress? These controls would help address that question and strengthen the conclusions of the paper.

For the control group, cells were kept for 60 min following isolation prior to fixation for imaging. This 60 min time point then matched the duration of the longest ISO exposure, ensuring that there was not time-dependent run down of cardiomyocyte quality that contributed to observed changes in RyR configuration. This has now been mentioned in the Results section on page2, line 142.

3) There is no indication of statistical significance on any of the summary data plots or mention of P values. This should be addressed.

Significant differences are now indicated in each plot by the use of comparison bars. Significance was assigned for P < 0.05, as noted in each figure legend, and the Methods section.

4) The modeling results need to be put into context and compared with what is going on in the experimental data. This section is somewhat confusing and is a bit of a jarring shift away from the experimental section. It should be better integrated. The authors state "the modeling results indicate that RyR ca^2+^ leak shifts from spark-mediated to silent leak when CRUs are dispersed, reducing the fidelity of spark generation" but in figure 4h, spark-mediated (non-silent) leak is strikingly increased after 60 mins of ISO treatment which leads to the most profound dispersal. Please clarify how these results align. Perhaps there is more of both spark-mediated leak and non-spark leak? What experimental data supports non-spark mediated leak? The idea that there could be localized regions of SR ca^2+^ depletion that act as "fire-breaks" to prevent the generation of ca^2+^ waves even though there are more ca^2+^ sparks is logical and provides a satisfying theory that unifies some of the seemingly paradoxical findings but this is only raised in the second to last paragraph of the discussion. Raising this idea earlier may help the reader connect the dots more easily.

Thank you for this important critique. We agree that in the original version of the article, it was difficult to understand the link between experimental and modeling data. We have added a paragraph to address this on page 3, line 282:

“To link observed RyR cluster morphologies with measured spark properties we constructed spark models employing experimentally-observed CRU morphologies. We then ran the computational spark model with these geometries to examine the resulting sparks. We also examined the likelihood of “undetectable” local release events whose amplitude is too small to be detected in confocal images. While difficult to measure experimentally, this non-spark mediated leak can importantly affect local SR load.”

In addition, we have reorganized this section considerably, changing the order of data presentation, and have repeatedly referred back to previous experimental findings to explain how they are paralleled by modeling results. The Reviewer correctly points out that data concerning spark-mediated and silent leak were not clearly presented. In the revised text, we have clarified that during β-AR, the modeling predicts that both types of leak are increased, which would promote reduction in SR content, and inhibition of ca^2+^ waves. We do not have experimental measurements of silent leak, so this insight from modeling is quite important. As suggested, we have now included mention of the “fire breaks” concept in the Results section where these data are first presented on page 6, line 340:

“Taken together, the modeling results critically link changes in CRU configuration and spark morphology; phenomena that were observed separately in experiments. However, the modeling data also support the contention from experimental data that RyR cluster fragmentation can reduce arrhythmogenic spontaneous ca^2+^ release (ca^2+^ waves). Here, the antiarrhythmic actions of RyR dispersion appear to be linked to a reduction in SR content resulting from an increase in both spark-mediated and silent leak. We postulate that local regions of the cell with dispersed CRUs and lowered SR content may act as a physical barrier or “fire break” to impede ca^2+^ wave propagation.”